# Longer growing seasons will not offset growth loss in drought-prone temperate forests of Central-Southeast Europe

Jan Tumajer [1] ✉, Jakub Kašpar [2], Jan Altman [3,4], Nela Altmanová[3,5], J. Julio Camarero [6], Emil Cienciala [7,8], Vojtěch Čada [4], Tomáš Čihák[9], Jiří Doležal[3,5], Pavel Fibich[3,5], Pavel Janda[4], Ryszard Kaczka[1], Tomáš Kolář[8,10], Jiří Lehejček[11], Jiří Mašek[1], Radim Matula[4], Kateřina Neudertová Hellebrandová[9], Lenka Plavcová[4], Michal Rybníček[8,10], Miloš Rydval [4], Rohan Shetti[12], Miroslav Svoboda [4], Martin Šenfeldr[10], Pavel Šamonil[2,10], Ivana Vašíčková[2], Monika Vejpustková [9] & Václav Treml [1]

The radial growth of temperate forests responds to climate change with remarkable variation across space and between species. However, there is limited understanding of how growing season extension and increasing drought stress contribute to long-term growth trends. Here, we calibrate the VS-Lite growth model using 2013 tree-ring chronologies from ten broadleaved and five coniferous genera in Central-Southeast Europe to predict intra-annual wood formation under four SSP climate scenarios through the 21st century. Results show that forecasted summer drought stress will be temporarily offset by an extended growing season, leading to stable or positive trends in tree-ring widths until a tipping point in the 2040s–2050s. During the second half of the 21st century, high-emission scenarios lead to growth acceleration in humid coniferous forests due to growing season extension and enhanced growth rate. In contrast, forecasted extension of the growing season is insufficient to compensate for declining summer growth rates at drier sites, resulting in significant growth reduction for all genera, particularly during dry years. Our results demonstrate that adjusting intra-annual wood formation to seasonal moisture availability may become crucial for tree survival in warmer climates. Furthermore, we highlight that only low-emission scenarios support non-declining stem growth in dry forests with current species composition.

The radial growth of trees is a prerequisite for ecosystem services provided by forests such as timber, soil nutrients released from deadwood, microclimate regulation, and mitigation of climate change[1]. Indeed, understanding the growth dynamics of tree species becomes essential in the face of accelerated climate warming leading to the global redistribution of climate-growth response patterns. In cold-limited regions, trees show reduced temperature sensitivity

in wood formation[2,3] which might potentially promote the expansion of forests into areas previously too cold to host trees[4]. By contrast, the amplification of drought stress at the rear edge of the distribution might initiate a growth decline, posing a risk to the long-term survival of tree species due to increased drought-induced mortality[5–7]. To better predict future forest growth, it is essential to enhance our understanding of climate-growth responses across

environmental gradients and how these responses shift with warming[8].

Climate change influences wood formation by altering its timing (i.e., phenology) and rate (i.e., kinetics) during the year. In cold biomes, the wood formation is temperature-controlled, resulting in unimodal seasonal growth patterns with a single growth peak in spring or summer and long winter dormancy[9]. Accordingly, the prolongation of the growing season and higher growth rates were frequently observed in response to increasing temperatures in boreal and mountain forests[10]. The response of growth rate and duration to warming might be complex in seasonally dry environments with long growing seasons, short winter dormancy, and facultative summer quiescence due to drought stress[11]. Divergent species-specific growth trends spanning from growth acceleration to steep declines have recently been reported in temperate biomes where species with different intra-annual growth patterns coexist[12–15]. For instance, ring-porous broadleaves often exhibit long growing seasons which may lack a distinct growth peak, while the growth of conifers and diffuse-porous broadleaves typically culminates in summer[16,17]. Notably, the effect of changing spring temperature on temperate growth phenology and how it translates into tree-ring width is not linear due to the interplay between chilling and forcing temperatures driving spring cambial reactivation[18,19], and increasing risks of late frosts with climate warming[20]. Furthermore, the site and species-specific growth phenology interacts with meteorological extremes, such as droughts and frosts, whose effects on tree growth and fitness depend on their timing within the growing season[20,21]. Given this complexity, attributing contributions of growth phenology and kinetics to the overall growth trends of temperate forests remains challenging[16,22,23]. For instance, an earlier start of the growing season might result in no change or even narrower tree rings of temperate tree species[24–26]. A better understanding of the effects of phenology and growth kinetics on long-term forest productivity across temperate species and regions can inform needed forestry adaptations to climate change.

Monitoring of stem phenology and kinetics with sub-annual resolution can be performed using dendrometers[27] and xylogenesis sampling[28]. However, existing datasets are scarce and often too short to represent interspecific variation in growth responses to climate change on a regional scale. Regional variation in stem growth phenology can be approximated by spectral indices of the canopy which often show acceptable agreement mainly for spring phenophases[29]. Moreover, process-based or empirical models of wood formation can overcome the lack of long dendrometer and xylogenesis records by simulating phenology and intra-annual growth rate from environmental proxies[30]. Among these models, VS-Lite[31] is an empirical nonlinear climate-driven model of tree-ring formation simulating dimensionless proxy for growth rate based on monthly temperature, soil moisture, and photoperiod. Since the model can be calibrated using empirical series of annual growth rates, it has been readily used for large networks of sites with available tree-ring data[6,32,33]. Yet, its outputs have not been validated against a comprehensive empirical dataset from dendrometers and only rarely they have been compared with other indirect proxies of growth phenology[34].

In this study, we use the VS-Lite model to forecast growth phenology and kinetics and their climatic drivers for the main temperate tree species in Central, Eastern, and Southeastern Europe toward the end of the 21st century (Fig. 1). We calibrate the model for 2013 tree-ring width chronologies of 15 tree genera distributed from drought-prone lowlands to Carpathian treelines. To independently validate simulated growth phenology and kinetics, we compare the simulations with dendrometer data available mostly for a single year at 57 sites, and with the proxy of the leaf phenology estimated from the Normalized Difference Vegetation Index (NDVI) for the 2000–2020 period at each site. Next, we run calibrated models with climatic projections based on the Shared Socioeconomic Pathways (SSP) with expected rates of

mean global temperature increase between 1.8 °C and 4.4 °C by the end of the 21st century[35]. To simulate the effects of continuous climate change and account for potential shifts in the intensity of meteorological extremes, we use SSP scenarios to define intra-annual climatologies representing mean, extremely warm-dry, and extremely cool-wet years expected through the 21st century (see the Methods section on how the climatologies were derived). We model growth phenology (i.e., start and end of growing season), monthly growth rates, total tree-ring width, and the main climatic growth-limiting factors for the calibration period 1961–2020 as well as for mean and extreme forecasted climatologies. We evaluate the forecasted shifts in growth phenology and kinetics, as well as overall growth trends, and assess their variability across environmental gradients and genera.

We hypothesized that (i) the future negative impacts of increasing summer drought stress on stem growth will be partly offset by the prolongation of the growing season. Moreover, we expected that the net balance between the positive effects of growing season extension and negative effects of summer growth decline would vary (ii) along the climatic gradient, (iii) among SSP scenarios, and (iv) during future meteorological extremes including cool-wet and warm-dry years. We show that the extension of the growing season will outweigh the negative impacts of reduced summer growth kinetics on tree-ring widths until the 2040s-2050s. During the second half of the 21st century, stable or positive growth trends are forecasted only in humid forests with cold-tolerant conifers provided that meteorological extremes remain rare. By contrast, predictions for drier sites, fast warming scenarios, and dry-warm extremes suggest significant reductions in tree-ring widths together with shifting intra-annual growth patterns from a short growing season with a single growth peak in summer to two distinct peaks in spring and autumn.

## Results
### VS-Lite performance
During the 1961–2020 calibration period, the mean correlation coefficient between observed and simulated tree-ring width chronologies was r = 0.42 (SE = 0.003). A significant Pearson's correlation was found at 84 % (p < 0.05, mean r = 0.46, mean 95% confidence interval = 0.21–0.64, mean $t_{(df)}$ = 3.78 with df between 33 and 58 depending on last year of calibration chronology) and 66 % (p < 0.01, mean r = 0.49, mean 95% confidence interval = 0.26–0.67, mean $t_{(df)}$ = 4.17) of 2,013 sites for which we calibrated the VS-Lite model (Fig. 2a). High mean correlations characterized *Pinus* sp., *Picea* sp., and *Quercus* sp. sites, while the lowest occurred in the case of *Fagus* sp. Simulated intra-annual growth patterns agreed with empirical data from dendrometers available mostly for a single year at 57 sites (Fig. 2b). The minor differences between dendrometer records and simulations occurred only in autumn when the model slightly overestimated relative growth rates compared to empirical observations. The coherence with dendrometer records was higher for conifers and *Fagus* sp. compared to ring-porous broadleaves (Supplementary Figs. 1-2). Dates of the mean NDVI crossing 0.5 of the annual amplitude coincided with spring increases and autumn declines in simulated growth rates for canopy-forming genera like *Fagus* sp., *Quercus* sp., and *Picea* sp. (Fig. 2c, Supplementary Fig. 3). Decoupling between NDVI and the VS-Lite simulations occurred mainly for genera of sparse stands (*Alnus* sp., *Pinus* sp.). Bootstrapped transfer functions showed a high level of temporal stationarity of VS-Lite forecasts with 84 % of sites having significantly stationary trends (Supplementary Fig. 4). The distribution of calibrated values of model parameters is provided in Supplementary Fig. 5.

### Growth forecast under climate change projections
By 2040–2059, predicted annual growth increments under mean climatology, i.e., forecast bi-decadal averages of temperature and precipitation, were larger than those observed during the 1961–2020

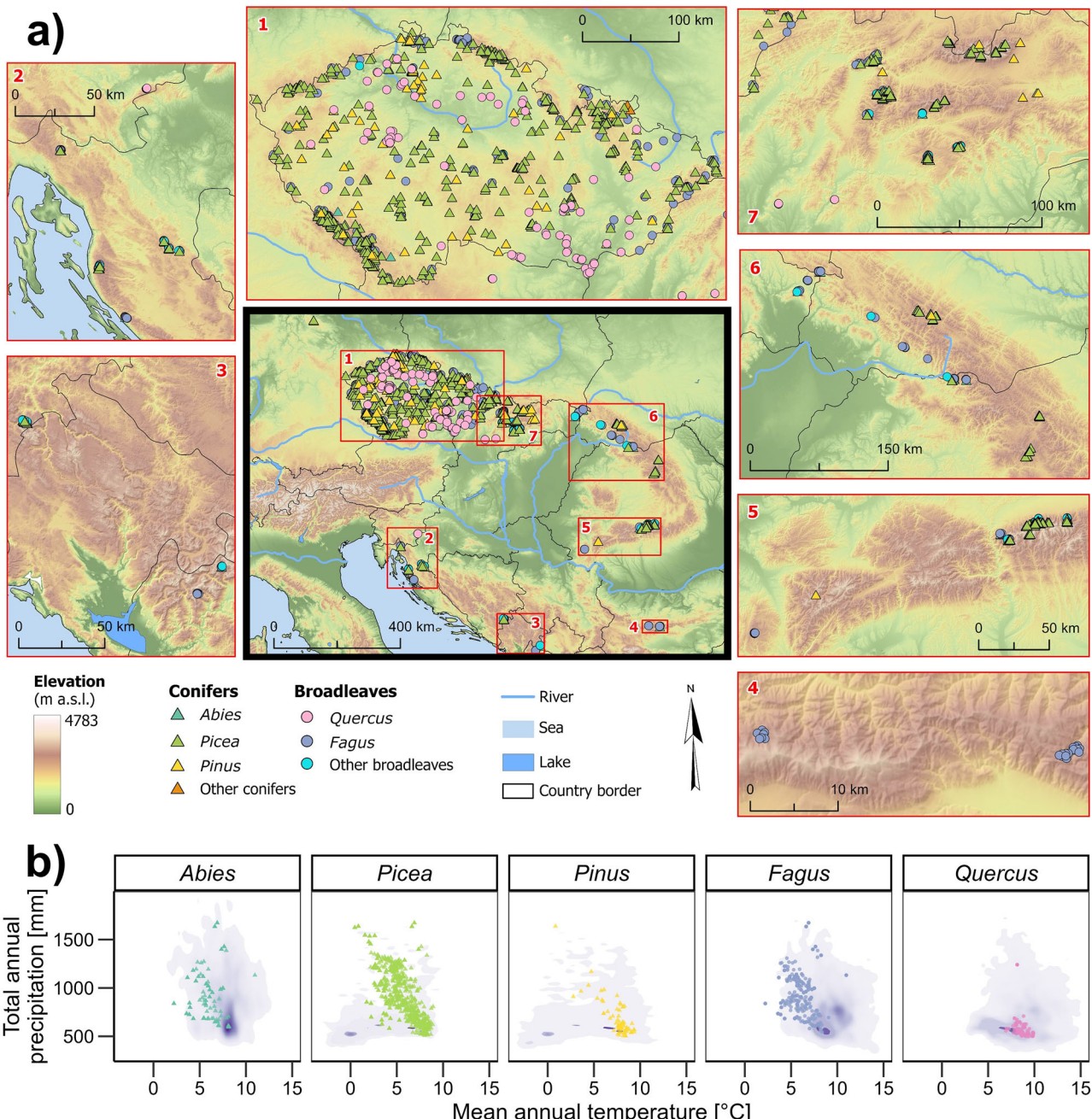

**Fig. 1 | Distribution of 2,013 sampling sites. a** Map of all sampling sites across Central, Eastern, and Southeastern Europe, **b** scatterplot of mean annual air temperature and annual total precipitation during 1961–2020 period at sites of the main tree genera. Purple polygons in (**b**) represent two-dimensional density charts of the climatic conditions across current European distribution of the key species of given genera (*Abies alba, Picea abies, Pinus sylvestris, Fagus sylvatica, Quercus robur,* and *Q. petraea*) based on chorological maps available from (https://data.mendeley.com/datasets/hr5h2hcgg4/6). The underlying map in (**a**) was created using the Natural Earth database and SRTM digital elevation model (NASA/USGS). Source data for (**b**) are provided as a Source Data file.

baseline period for most SSP scenarios and genera (Fig. 3, Supplementary Fig. 6). By contrast, a prominent divergence between scenarios was predicted for the second half of the century. Under the low-emission scenario SSP1-2.6, average tree-ring widths in the 2080–2099 period were predicted to be on average 15 %, 13 %, and 28 % wider compared to the baseline period at dry (from -160 to +200 mm relative to the annual climatic water balance, i.e., the difference between annual precipitation and potential evapotranspiration), moderate (200–700 mm), and humid (700–1230 mm) sites, respectively. The simulated annual ring width at the end of the century under the low emission SSP2-4.5 scenario increased at humid (35 % increase

compared to the baseline period) and moderate sites (12 %), but an increase in mean tree-ring width was negligible at dry sites (4 %). Simulated tree-ring widths in humid environments were substantially larger at the end of the 21st century compared to the baseline period under high-emission SSP3-7.0 and SSP5-8.5 scenarios (42 % and 48 %). At moderate sites, high-emission scenarios produced mean tree-ring widths similar to the baseline period. By contrast, mean annual increments of dry sites were predicted to be on average 13 and 27 % narrower at the end of the 21st century compared to the baseline period under mean climatology of the SSP3-7.0 and SSP5-8.5 scenarios, respectively. Both positive and negative differences in mean tree-ring

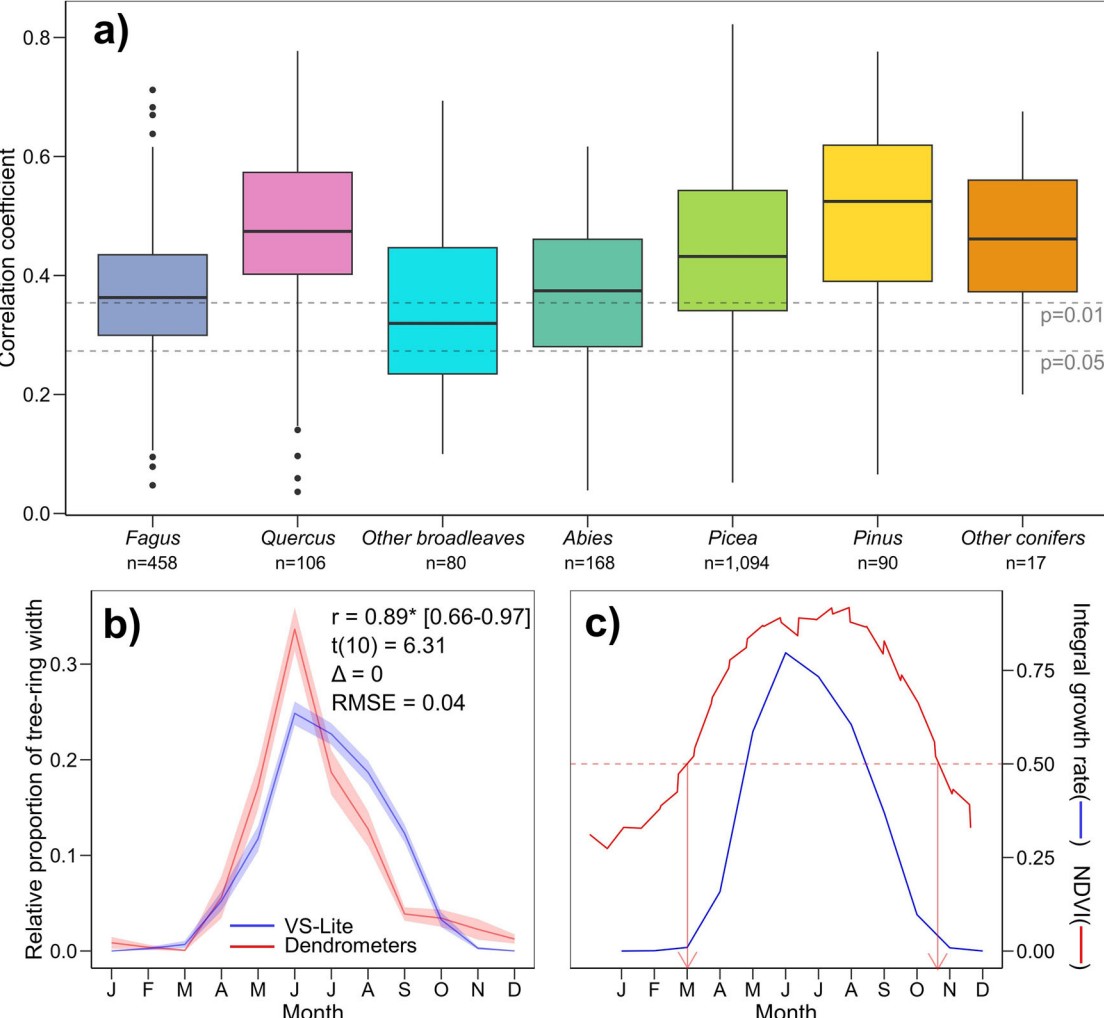

**Fig. 2 | Coherence between simulated growth rates produced by the VS-Lite model and empirical datasets. a** Boxplots of correlation coefficients between simulated and observed tree-ring width chronologies during the calibration period 1961–2020, **b** mean monthly rate of tree-ring formation simulated by the model (blue line) and recorded by dendrometers (red line) at 57 sites and 83 individual years within the 2012–2020 period standardized by final simulated or measured tree-ring width, and (**c**) mean intra-annual variability of simulated integral growth rates (blue line) and the standardized Normalized Difference Vegetation Index (NDVI, red line) for the period 2000–2020. Boxplots in (**a**) show median (horizontal line), 25th and 75th percentile (hinges), most extreme observations no further than 1.5x interquartile range from the hinge (whiskers), and outliers (points). n in (**a**) represents replication of sites per genera and vertical dashed lines highlight

significance levels for two-sided tests on Persons' correlation coefficient (degrees of freedom = 50) without adjustment for multiple testing. Buffers around lines in (**b**) represent a 95% confidence interval around the mean and the values in the top of the chart indicate goodnes-of-fit statistics between dendrometer data and the simulations (r = Pearson correlation coefficient [its 95 % confidence interval], * = statistically significant correlation according to two-sided test with $p = 8.7 \times 10^{-5}$, t(10) = t-value for test with 10 degrees of freedom, *RMSE* root mean squared error, Δ = temporal offset in months of seasonal growth peaks). The red arrows in (**c**) highlight the mean date when NDVI exceeded or dropped below 0.5 of its annual amplitude. For charts (**b**, **c**) on the level of individual genera and sites, see Supplementary Figs. 1–3. Source data are provided as a Source Data file.

widths between mean climatology forecasts and the baseline period were statistically significant ($p < 0.05$) for most sites (mean difference in annul sum of integral growth rates = -0.13, mean 95% confidence interval = -0.28–0.00, mean $t_{(59)}$ = 0.88), mainly towards the end of the 21st century and high-emission scenarios (Supplementary Fig. 7).

The growth deficit, i.e., the proportion of the tree ring that was not formed due to climatic limitation, was mainly due to low temperatures across the entire gradient of climatic water balance during the baseline period (Fig. 4a, b). However, the growth deficit due to cold limitation was predicted to decline over time at the expense of growth limitation due to drought stress at a pace reflecting the warming rate of SSP scenarios (Supplementary Figs. 8-9). The mean simulated duration of the growing season during the baseline period shifted with climatic water balance from approximately three months at humid sites mainly at high elevations to almost six months at the driest sites (Fig. 4c, d).

The mean forecast extension of the growing season across all sites toward the end of the 21st century varied between 0.65 (SSP1-2.6) and 1.44 months (SSP5-8.5). The extension was driven mainly by an earlier start of cambial activity, while the end of the growing season showed site-specific shifts over time, largely depending on climatic water balance. Although delayed growth cessation was mostly predicted at moderate and humid sites due to warming autumn temperatures, it frequently occurred earlier in dry sites due to amplified drought stress causing growth cessation in the late summer (Supplementary Fig. 10).

Simulated integral growth rates for the baseline period had a prevailingly unimodal intra-annual distribution with a single growth peak around the summer solstice across the entire climatic water balance gradient (Fig. 5, Supplementary Fig. 11). For mean climatology based on low-emission scenarios, the intra-annual growth pattern was predicted to remain unimodal with weak summer growth reductions at

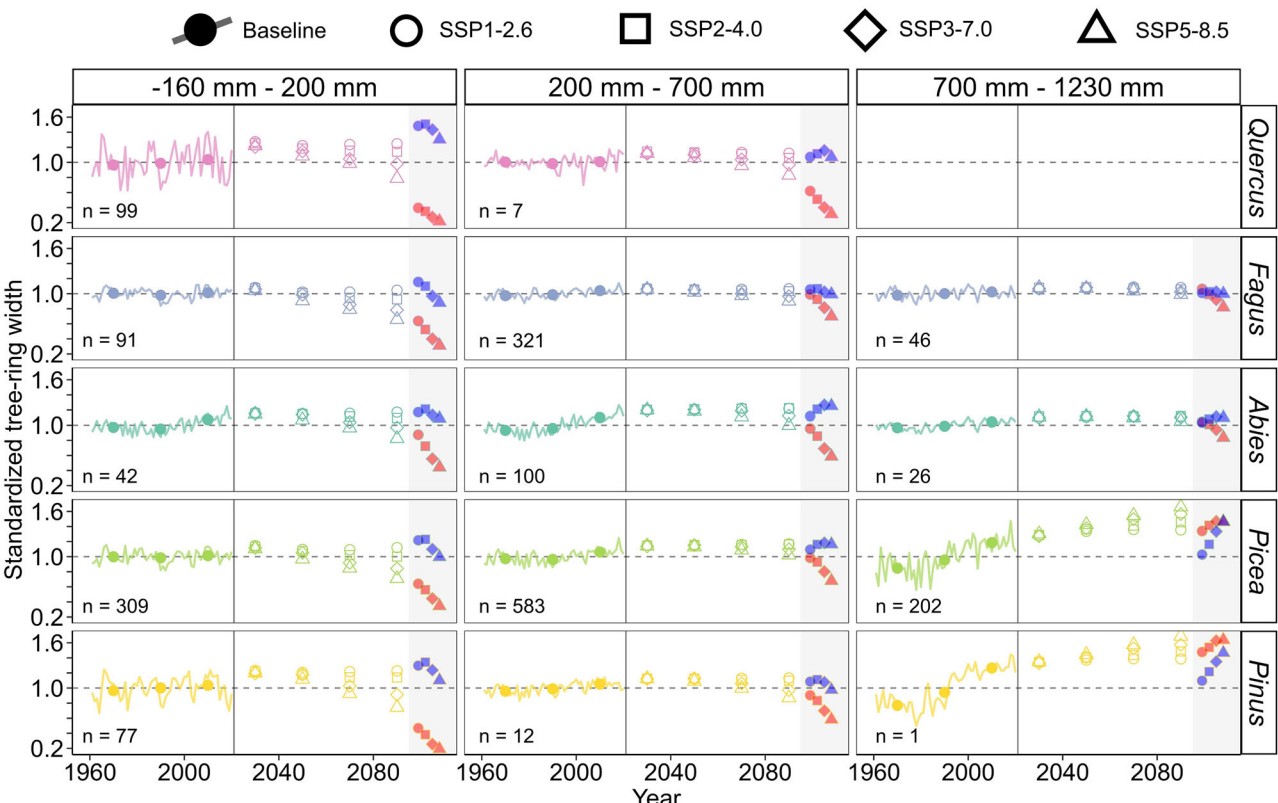

**Fig. 3 | Simulated annual growth increments (i.e., tree-ring width indices).**
Simulated increments are shown for a baseline calibration period 1961–2020 (lines for annual values, full symbols for bi-decadal means) and forecasts for mean climatic conditions during four bi-decadal periods between 2020–2039 and 2080–2099 based on four SSP scenarios of climate change (empty symbols for bi-decadal means) for the five most common genera. Full symbols in shaded areas on the right of each plot represent simulated annual growth increments during meteorological extremes in the 2080–2099 period including cool-wet (blue) and warm-dry years (red). Sites were averaged into three clusters according to mean annual climatic water balance in the baseline period including dry (-160–200 mm), moderate (200–700 mm), and humid (700–1230 mm). n indicates the number of sites for a given range of climatic water balance and genus. Tree-ring width indices were standardized by mean simulated tree-ring widths in the baseline period. For other genera see Supplementary Fig. 6. Source data are provided as a Source Data file.

dry sites during the second half of the 21st century. By contrast, forecast growth rates under high-emission scenarios significantly accelerated ($p < 0.05$) in spring and autumn (e.g., mean test statistics for April and May: mean difference in growth rates = 0.11, mean 95% confidence interval = 0.08–0.15, mean $t_{(59)}$ = -3.13) but significantly declined in summer (e.g., mean test statistics for June-August: mean difference in growth rates = -0.15, mean 95% confidence interval = -0.19–-0.11, mean $t_{(59)}$ = 7.79, Supplementary Fig. 12). Consequently, the pattern of integral growth rates tended to shift into right-skewed unimodal or even bimodal form under high-emission SSP scenarios (Supplementary Fig. 13). The nearly symmetric unimodal pattern persisted till the end of the 21st century only at humid sites of *Picea* sp. and *Pinus* sp., however, the latter genus being underrepresented in humid sites in our dataset.

**Growth forecast under extreme climatology**
Running calibrated VS-Lite models using extreme climatologies, i.e., forecast monthly temperature and precipitation for unusually cool-wet and warm-dry years within each bi-decadal period, resulted in amplified shifts of both annual and intra-annual growth from the baseline mean. Notably, the highest differences between predictions based on extreme and mean climatologies were at dry sites and diminished with increasing climatic water balance. Simulations for the 2080–2099 period suggested mean tree-ring width reductions at dry sites during warm-dry years between 41 % (SSP1-2.6) and 70 % (SSP5-8.5) compared with the 1961–2020 baseline mean (Fig. 3). For cool-wet years, simulations across all scenarios indicated wider tree rings at dry

sites by the end of the century compared to both the baseline mean and the projected mean climatology. At humid sites, the mean forecast tree-ring widths during warm-dry years of the 2080–2099 period were 26–31 % larger compared with the baseline mean but lower than forecasts based on mean climatology. Predicted growth during cool-wet years at humid sites was lower compared with warm-dry years, mainly for low-emission SSP scenarios. Differences between baseline and forecast tree-ring widths under both types of extreme climate were statistically significant at most sites regardless SSP scenario (Supplementary Fig. 7).

The growing season was predicted to start earlier during warm-dry years and later during cool-wet years compared to mean climatology (Fig. 4c, d). Results suggested a widespread shift toward earlier cessation of the growing season during warm-dry years in the second half of the 21st century, particularly at sites with low climatic water balance and high-emission scenarios. This was due to increasing drought-driven growth deficits and increased frequency of zero growth rates simulated in summer and autumn during warm-dry years (Fig. 4a, b; Supplementary Fig. 10). The summer growth deficit was partly alleviated during cool-wet years compared to the mean climatology (Fig. 5, Supplementary Fig. 12).

## Discussion
The simulations highlighted a trade-off between the extension of the growing season and summer drought stress toward the end of the 21st century and their net effects on annual growth increments of temperate tree species in Central, Eastern, and Southeastern Europe. Current

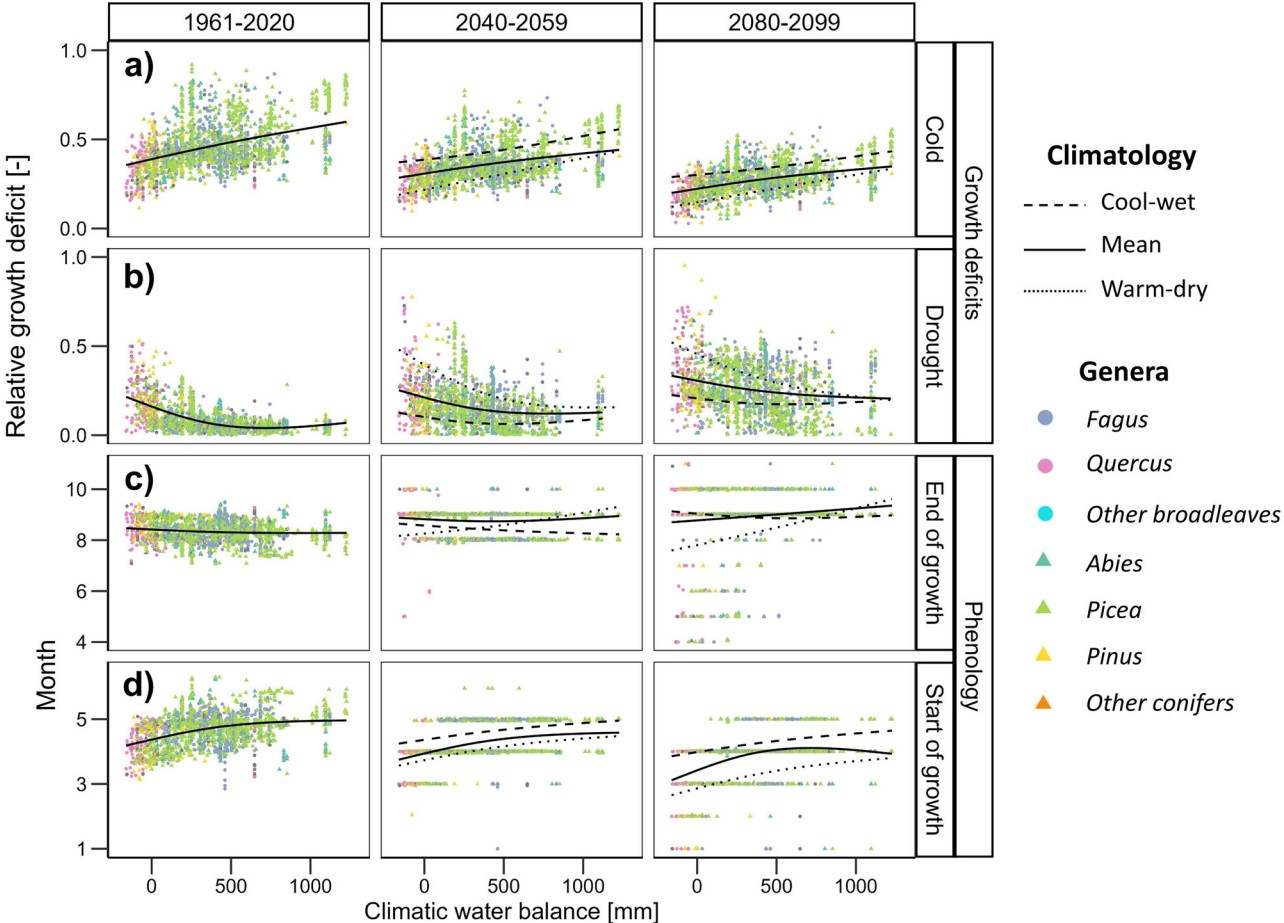

**Fig. 4 | Simulated growth deficits and phenology.** Mean simulated growth deficits due to (**a**) cold and (**b**) drought stress, and timing of growing season (**c**) end and (**d**) start for a baseline period 1961–2020, and forecast for 2040–2059 and 2080–2099 based on the high-emission SSP5-8.5 scenario. Points show mean values for each site, for forecasts they are based on the mean climatology of given bi-decadal period. Solid lines represent generalized additive models fitted through individual site observations for mean climatology along the gradient of climatic water balance. Dashed and dotted lines highlight a shift of generalized additive models if forecasts for climatic extremes within a given bi-decadal period were considered instead of the mean climatology. For other SSP scenarios see Supplementary Figs. 8–9. Source data are provided as a Source Data file.

cambial activity in the region is limited mainly by low temperatures as indicated by prevailing cold-driven growth deficits. All forecasts agreed on the alleviation of cold limitation as climate change progresses, though with divergent growth trajectories based on rates of growing season extension and summer drought stress. Tree growth might become increasingly limited by the reduced summer growth rate due to amplified drought, mainly in exceptionally warm-dry years. These negative impacts might be offset or even outweighed by the extension of the growing season due to earlier cambial reactivation and higher growth rates in spring and autumn. Consequently, the simulations forecast a gradual transition from the unimodal growth pattern into a long growing season with right-skewed or even bimodal intra-annual growth dynamics at dry sites. We observed systematic variation in the net effects of shifting kinetics and phenology along the gradient of water availability, primarily reflecting the pace and seasonality of warming throughout the 21st century and the occurrence of stochastic warm-dry climatic extremes.

## Forecast annual growth rates

Simulated annual growth rates showed a high similarity between low-emission (SSP1-2.6, SSP2-4.5) and high-emission (SSP3-7.0, SSP5-8.5) scenarios of mean climatology until 2040–2059. Mean predicted tree-ring width indices during this period remained stable or slightly increased compared to the 1961–2020 baseline period. They were

similar to values from 2005–2010, i.e., relatively wetter years compared to the most recent decade (2010s) characterized by amplified drought stress[36]. This suggests that temperate forests might keep benefiting from prevailingly increasing growth trends[12,14,15] for the next few decades, if climatic extremes, including drought spells, are absent or occur only rarely.

Forecasts based on mean climatology from high and low-emission scenarios diverged during the second half of the 21st century. While low-emission scenarios predicted annual growth rates fluctuating close to the baseline mean, the high-emission scenarios associated with a rapid temperature increase produced unprecedented trends in simulated tree-ring widths. Notably, both conifers and broadleaves at the dry edge of climatic space captured in our study are expected to substantially reduce growth compared to the baseline under the SSP5-8.5 and SSP3-7.0 scenarios. The forecast reductions in tree-ring widths are larger than the mean growth declines simulated for the same species using a similar framework in dry forests in Northeastern Spain[6]. This suggests that forests from temperate Europe might be more vulnerable to drought stress compared to the Mediterranean due to lower resilience after drought events[37].

In contrast to growth declines simulated at dry sites, humid edges of coniferous species distribution, mainly of *Picea* sp., are expected to profit from high-emission scenarios of climate warming. By the end of

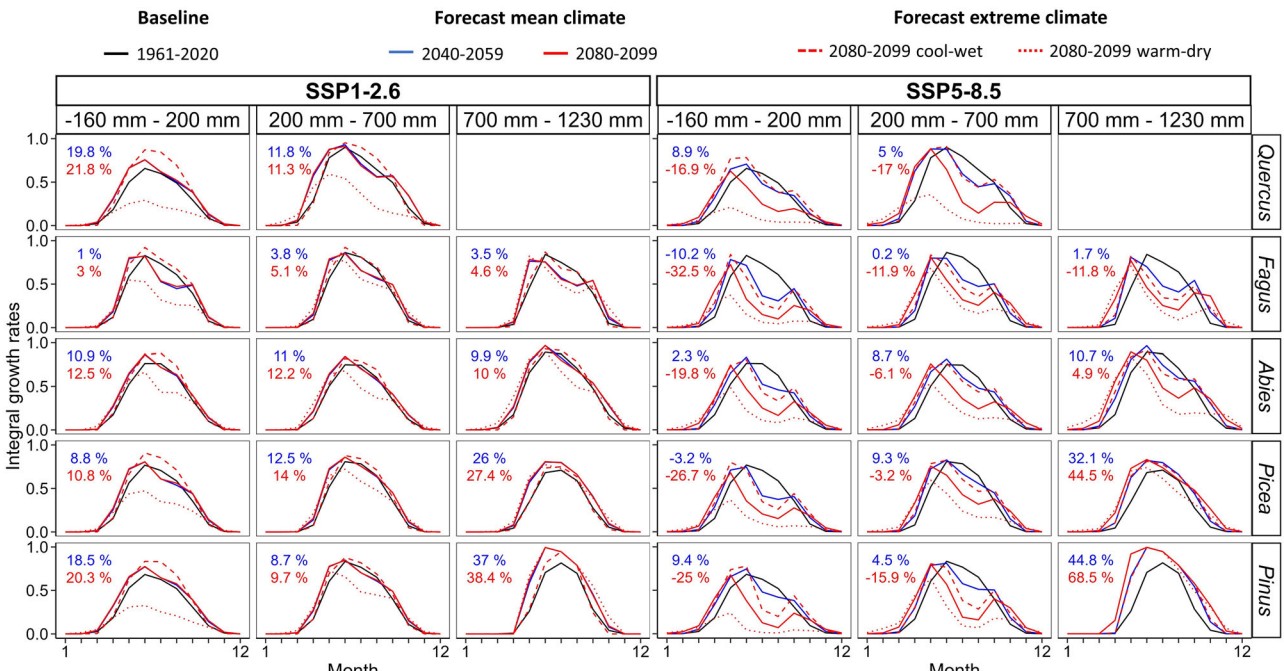

**Fig. 5 | Simulated intra-annual growth patterns.** Mean simulated monthly integral growth rates for the baseline period 1961–2020 (black), 2040–2059 (blue), and 2080–2099 (red) for the five most common genera along a gradient of climatic water balance reflecting low-emission SSP1-2.6 (left) and high-emission SSP5-8.5 (right) scenarios. Red lines are based on simulations reflecting mean (solid), extremely cool-wet (dashed), and extremely warm-dry (dotted) years predicted for the 2080–2099 period. Sites were averaged into groups according to mean annual climatic water balance in the baseline period including dry (-160–200 mm), moderate (200–700 mm), and humid (700–1230 mm). Values in the top-left corner indicate the difference between the annual sum of integral growth rates between forecasts for future decades under mean climatology and the baseline period. For other genera and SSP scenarios, see Supplementary Fig. 11. Source data are provided as a Source Data file.

the century, the mean annual growth of these mostly high-elevation stands might accelerate for almost 50 % of the baseline mean if climate follows the SSP5-8.5 scenario and mean climatology. This is in line with currently observed positive growth trends and an increase in the productivity of cold ecosystems across the globe[4,10]. Such intensification of the growth rate might challenge mountain forestry in Central-Southeast Europe by accelerating stand dynamics and tree turnover, due to either a negative feedback between tree growth rate and life-span or a higher sensitivity of fast-growing trees to disturbances[38]. Consequently, faster growth can, counterintuitively, lead to shortened carbon residence times in mountain forests[39]. Indeed, high-emission SSP scenarios might cause continuously declining forest growth at dry sites and benefit humid forests with fast turnover. The diverging growth trajectories across the landscape might pose a significant challenge to future forest and landscape management.

Predicted annual growth rates during future climatic extremes revealed more severe growth loss at dry sites under warm-dry spells compared to mean climatology. Notably, simulated tree-ring widths were between 41% and 70% narrower compared to baseline mean during warm-dry years of the 2080–2099 period at sites with a current climatic water balance of less than 200 mm per year. These results are alarming considering previous empirical observations of forest die-back triggered by reduced growth during dry spells in Palearctic temperate forests[5,7,40]. Integrating tree-ring formation models, such as VS-Lite, with models simulating stand processes is essential for understanding how projected growth reduction contributes to the risk of tree die-off[41]. Although cool-wet years may alleviate drought stress and increase growth at dry sites during the 2080–2099 period, forecasts for both warm-dry and cool-wet extremes led to reduced growth at humid sites. This illustrates, that humid sites, i.e., mostly mountain forests, might show future sensitivity to both summer droughts and cool growing seasons.

## Forecast phenology and growth kinetics

Annual growth rates represent a product of growing season duration and rate of wood formation throughout the year[42]. Increasing air temperature usually stimulates the growing season duration but indirectly limits summer growth kinetics through the moisture availability[43]. Consequently, the forecast growth trends primarily reflected the capacity of the growing season extension to offset reduced summer growth rates. According to simulations for the 1961–2020 baseline period, the mean duration of the growing season currently varies between approximately three months at humid sites and six months at dry sites. Moreover, most of the simulated growth deficits were due to cold rather than drought stress, and the simulated growth cessation mostly occurred due to cold rather than drought-limitation (Supplementary Fig. 10). This shows that climate warming might promote an extension of the growing season to some extent under warmer but not drier climates. Accordingly, simulating the earlier start of the growing season outweighed the aggravating summer drought stress at most sites regardless of the SSP scenario considering the mean climatology of the 2040–2059 period. The mechanistic representation of growth phenology in the VS-Lite model might be a reason for slightly better growth trends predicted till the 2050s in our study compared with statistical forecasts based on seasonal climatic means[8,13].

From 2040–2059 onwards, the net effect of the growing season extension and drought-driven reduction of growth kinetics depended on the rate of climate change. While the simulated intra-annual growth patterns for mean climatology under the SSP1-2.6 scenario were mostly stable during the second half of the 21st century, they showed systematic shifts under high-emission forecasts. Scenario SSP5-8.5 predicted growing seasons to be on average 1.44 months longer at the end of the century compared to the baseline period, mainly due to earlier spring growth onset (Fig. 4c, d). However, simulations suggest

that such a pace of growing season extension and associated acceleration of growth in spring and autumn will not be sufficient to compensate for steep summer reductions of growth kinetics at dry sites[22]. Accordingly, the intra-annual growth dynamics predicted for the mean climatology at the end of the century in low and middle elevations were characterized by a long growing season but low growth rates, with a local minimum in summer and potentially two growth peaks in spring and autumn. The phenomenon of right-skewed or bimodal growth over the year is characteristic for seasonally dry environments and might represent an adaptation to temporarily withstand summer drought that is typical of Mediterranean climates[11,44]. However, autumn growth reactivation leading to bimodality is known to be facultative, i.e., it might occur irregularly in time and across space depending on precipitation seasonality and growth plasticity of a given species[45,46]. Notably, the autumn growth peak at dry sites was diminished in simulations for warm-dry years at the end of the 21st century and high-emission scenarios. As a consequence, predicted dates for the end of the growing season were ambiguous along the humidity gradient due to the frequent fading of the autumn growth peak at dry sites. This highlights the limited ability of phenological shifts and accelerated autumn growth to offset drought stress in a long-term.

The VS-Lite model simulated annual and monthly growth increments with a high coherence with empirical datasets of tree-ring width chronologies, dendrometer data, and NDVI series of canopy-forming species in the baseline period. This confirms its capability to approximate climatic drivers of stem growth, cambial phenology, and canopy greenness across large tree-ring networks[6,32–34]. However, how tightly the growth will agree with the model in the forecasting period might depend on species-specific growth plasticity, i.e., the physiological ability to shift from a unimodal growth pattern with a short growing season common in the baseline period into longer growth with multiple peaks predicted for the end of the century. Although most species considered in our study have temperate or boreal ranges with the prevalence of unimodal growth, observed growth patterns have demonstrated their capability to reduce growth in summer and accelerate in autumn under seasonally dry climates. For instance, growth bimodality has been reported during recent dry years using dendrometers or xylogenesis monitoring primarily for *Pinus sylvestris*[47] and *Picea abies*[26,48]. Moreover, all major conifers including *P. sylvestris*[49], *P. abies*[50], and *Abies alba*[51] frequently form intra-annual density fluctuations within the latewood, which are deemed to evidence an acceleration of autumn growth in response to precipitation after a dry summer. Available dendrometer data from Central Europe show common growth multimodality for ring-porous broadleaves including *Quercus robur*, *Quercus petraea*, and *Fraxinus excelsior* although the growth of diffuse-porous *Fagus sylvatica* is rather unimodal due to photoperiod constraints of its phenology[29]. Accordingly, a significant proportion of temperate species, particularly widespread conifers and oaks but not diffuse-porous broadleaves, should be capable of tracking predicted bimodal growth pattern. The growth seasonality might, therefore, become an important competitive trait under warmer and drier climate.

## Model and database limitations
Our study is based on a new regional dataset of 2013 tree-ring width chronologies. Although the majority of these sites are within the core of the current climatic niche of the investigated genera in Europe, the coldest (e.g., Scandinavian coniferous stands) and driest (e.g., Western Mediterranean oak stands) continental margins are not represented (Fig. 1b), which restricts the generalization of our findings to the temperate zone of Central, Eastern, and Southeastern Europe. Correlations between observed and simulated chronologies peaked for conifers and *Quercus* sp. but were the lowest for diffuse-porous broadleaves like *Fagus* sp. This might reflect the spatial distribution of individual sites in our study region, where conifers and *Quercus* sp. often occupy climatic margins of forest distribution, including treelines and dry lowlands, while *Fagus* sp. forms forests in medium elevations. Moreover, better performance for coniferous sites might be a legacy of the VS-Lite and Vaganov-Shashkin models originally designed for simulating the growth of boreal conifers[32].

Numerical growth forecasts assume temporal stationarity of climate-growth interactions calibrated in the baseline period. The stationarity of the VS-Lite model was shown to be high for major Palearctic conifer species[52] and this was confirmed for our diverse dataset using an independent trial (Supplementary Fig. 4). Our predictions capture robust long-term growth trends expected till the end of the 21st century but ignore year-to-year variation beyond the mean and extreme climatologies for each bi-decadal period. By restricting our growth forecasts to mean, cool-wet, and warm-dry years we aimed to reduce the effects of inter-annual uncertainty of climatic models driven by the stochastic nature of the climatic system.

The simulated phenological shifts might have been affected by the monthly temporal resolution of the VS-Lite model. For instance, autumn integral growth rates were slightly overestimated compared with dendrometer records (Fig. 2b). Wood formation models operating on daily temporal resolution might be better suited to simulate summer growth quiescence and short-term autumn reactivation under seasonally dry conditions[53]. However, the mean rate of growing season extension predicted by the end of the 21st century under high-emission scenarios, 0.39 days per year ($\approx$ 1.44 months over 110 years), is similar to currently observed phenological shifts in Central Europe[54], supporting the good performance of VS-Lite for estimating growth phenology[34]. Further studies should test improvements in phenological mechanisms of the VS-Lite model, including effects of winter conditions on growth phenology through chilling-forcing interactions[18], risks of late-frost damage after cambial reactivation[20], or non-climatic drivers of growth phenology[55]. Moreover, VS-Lite model might benefit from incorporating additional drivers of wood formation like vapor pressure deficit[43] which has recently been successfully implemented in the daily Vaganov-Shashkin model[56], although its predictive power for growth at a monthly scale still needs to be tested. Finally, the empirical model of tree-ring growth used in our study captured non-linear but fairly continuous shifts in phenology and intra-annual growth patterns in a response to changing climate. However, sites where climate change will trigger stochastic events including increased mortality might experience more dramatic and abrupt shifts of both intra-annual and inter-annual growth patterns[57]. Accordingly, our forecasts of intra-annual growth at the tree-ring level may be less reliable at sites affected by stand-replacing forest disturbances in the future[58].

## Implications of forecast phenological shifts for forest functioning
Shifting growth seasonality from summer to spring and autumn as predicted by our model for high-emission scenarios might have cascading impacts on forest functioning[59]. Empirical evidence suggests that the prolongation of the growing season stimulates spring carbon sequestration[60] but respiration outweighs increased assimilation in autumn[61]. In consequence, a significant extension of the growing season in temperate forests of the Northern Hemisphere together with reduced summer growth rates as predicted in our study might offset the phase and alter the amplitude of the annual cycle of atmospheric $CO_2$ concentrations[60]. Similar to the carbon sequestration, the seasonality of other processes linked to plant phenology and crucial for society including the water cycle, soil development, and landscape albedo might to some extent adjust to the forecast shifts in wood phenology[62]. Accordingly, forecasts of growth phenology and kinetics provided by our study contribute to understanding the broader implications of future shifts in the seasonality of temperate forests.

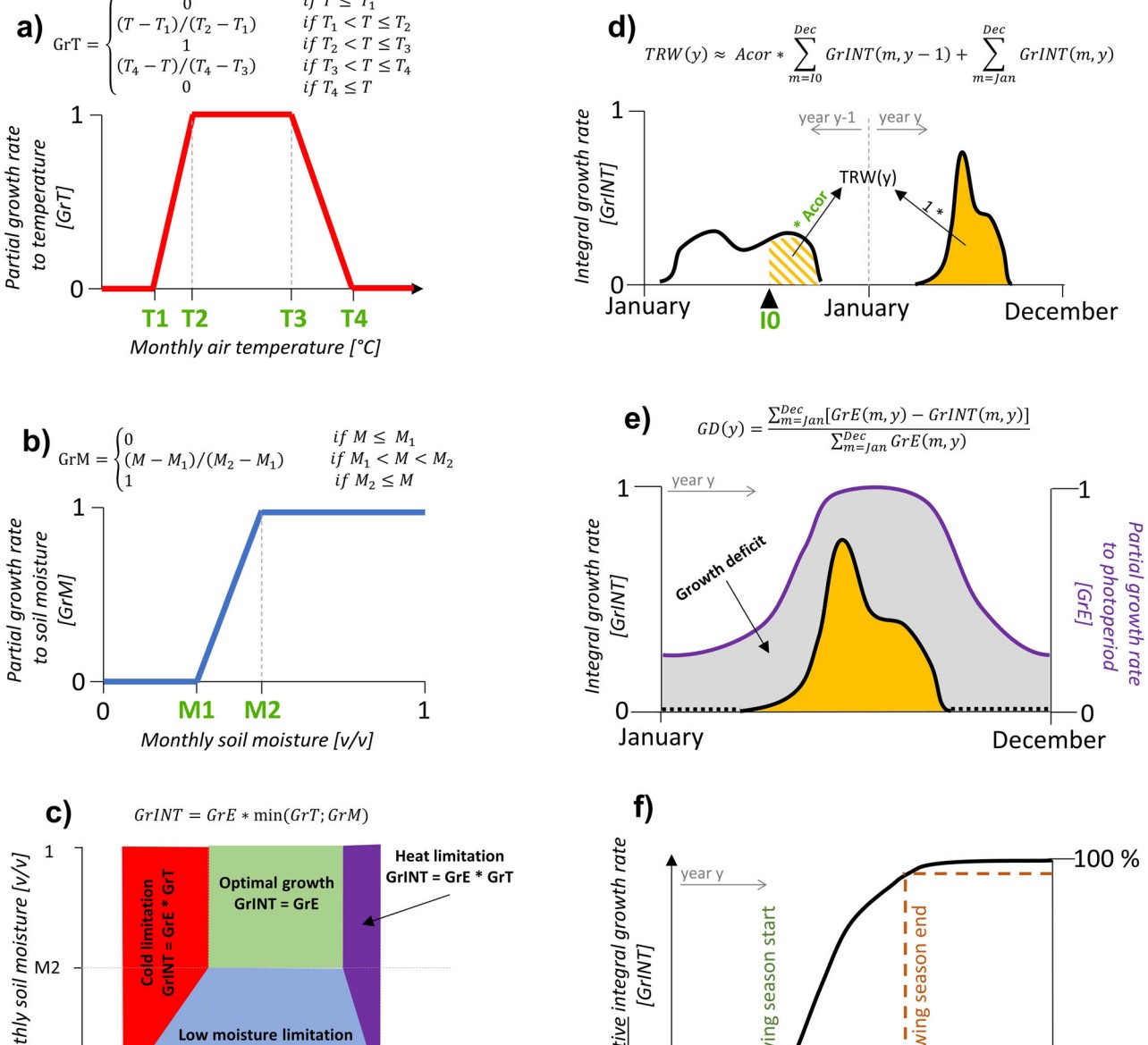

**Fig. 6 | Schematic and mathematical representation of the modified VS-Lite model workflow. a** Conversion of monthly mean air temperature (T) into dimensionless partial growth rate to temperature (GrT), **b** conversion of monthly mean relative volumetric soil moisture (M) into dimensionless partial growth rate to soil moisture (GrM), **c** calculating integral growth rate (GrINT) and determination of a dominant growth-limiting factor for each month based on monthly air temperature and soil moisture, **d** definition of a proxy for annual tree-ring width (TRW)

considering integral growth rates of two consecutive years, **e** estimating growth deficit (GD) as a difference between simulated integral growth rate (black line) and partial growth rate to photoperiod (GrE; purple line), **f** definition of the phenological phases based on intra-annual distribution of cumulative integral growth rates. Eight parameters of the model that have been calibrated for each site are highlighted in light-green bold letters. The dashed black line in (**e**) highlights months with growth cessation, i.e., GrINT=0. *y* year, *m* month.

## Methods

### Study area and tree-ring data

The calibration set integrated tree-ring width chronologies from three dendrochronological datasets from Central, Eastern, and Southeastern Europe. Specifically, we included tree-ring width series from the TreeDataClim database (www.treedataclim.cz/en)[63], the Czech landscape inventory CzechTerra[15], and the RemoteForests network of primary forests (www.remoteforests.org)[64]. While TreeDataClim stores tree-ring data collected mainly from dominant trees in managed or semi-natural forests with presumably strong climate-growth responses, CzechTerra employed a random sampling design, and Remote-Forests used stratified sampling focused on primary forests with limited forestry management but undergoing disturbance cycles. We used tree-ring width series of individual trees for sites with minimum replication of five trees over the entire 1961–1995 period to maximize both the length of the series and the density of sites. Altogether, our final dataset is represented by 2013 sites of 15 tree genera, including five genera of gymnosperms and ten genera of angiosperms (Fig. 1a).

The most common genera in the dataset were *Picea* sp. (1094 sites), *Fagus* sp. (458), *Abies* sp. (168), *Quercus* sp. (106), *Pinus* sp. (90), and *Acer* sp. (44; Supplementary Table 1). Our sites representatively cover mixed forests in Central, Eastern, and Southeastern Europe across prominent latitudinal (42.4–51.8°N), longitudinal (10.6–25.5°E), and elevational gradients (153–1713 m). The mean annual climatic water balance of individual sites defined as the difference between annual precipitation and potential evapotranspiration during the 1961–2020 period varies between -160 mm and +1230 mm. The dataset covers the central areas of the current climatic niche of the main genera but sites similar to dry-cool (*Picea* sp., *Pinus* sp.) or warm (*Fagus* sp., *Abies* sp.) distribution margins in Europe remain underrepresented (Fig. 1b). Sampling tree-ring cores complied with legal regulations and permissions for coring were obtained from respective forest owners and managers.

To remove ontogenetic trends from tree-ring widths we fitted a smoothing spline with a 50% frequency cut-off at 60 years to each series and divided observed tree-ring widths by the value of the spline[65]. The spline length was chosen to retain half of the variability within the calibration period (1961–2020) as a signal in the detrended series while removing the other half as an ontogenetic trend. The assumption that half of the variation in tree-ring widths is attributable to tree aging is justified by the average age of sampled trees, which is 128 years. Next, we averaged detrended series from the same site into standard tree-ring width chronologies using the robust biweight mean.

## Climatic data and climatic scenarios

We obtained monthly mean air temperature and total monthly precipitation for each site between January 1961 and December 2020. To do this, we used gridded climatic products available for the location of each site. For sites in the Czech Republic, we used spatially interpolated historical time series of temperature and precipitation based on data from local climatological stations of the Czech Hydrometeorological Institute and the German Weather Service[52]. The interpolation algorithms for temperature and precipitation were based on orographic regression and inverse-distance weighting, respectively. A search radius of 70 km (temperature) and 50 km (precipitation) was applied around each site to subset climatological stations used in the interpolation. For sites in other countries, we obtained monthly mean temperature and total monthly precipitation from the E-OBS[66] (version 30) gridded database with a spatial resolution of 0.1°×0.1°. Testing a subset of Czech sites revealed negligible differences in the results based on national and E-OBS climatic datasets.

In addition to historical climate, we retrieved mean monthly temperatures and monthly total precipitation for each site toward the end of 21st century based on a mean ensemble of bi-decadal anomalies estimated from the sixth Coupled Model Intercomparison Project (CMIP6). Mean anomalies in temperature and precipitation from the normal period 1995–2014 were downloaded for each calendar month with a spatial resolution of 0.25° × 0.25° through the Climate Change Knowledge Portal (https://climateknowledgeportal.worldbank.org/netcdf-browser?prefix=data/cmip6-x0.25/) for bi-decadal periods 2020–2039, 2040–2059, 2060–2079, and 2080–2099. We considered forecasts reflecting four SSPs, i.e., tailored scenarios of future emissions of greenhouse gases, mitigation and adaptation measures, and social development pathways (Supplementary Fig. 14; Supplementary Table 2). Scenario SSP1-2.6 anticipates a pervasive shift towards sustainable production and a gradual reduction of anthropogenic emissions. Contrary to SSP1-2.6, high-emission scenario SSP5-8.5 presumes rapid technological development and competitive markets leading to accelerated emissions of greenhouse gases. Moderate scenarios SSP2-4.5 and SSP3-7.0 were also included. Although the Intergovernmental Panel on Climate Change considered SSP2-4.5 the most likely scenario[67], current trends in emissions and global pollution policies seem to best align with SSP5-8.5[68]. Throughout the manuscript, we refer to SSP1-2.6 and SSP2-4.5 as 'low-emission' scenarios, and to SSP3-7.0 and SSP5-8.5 as 'high-emission' scenarios.

To calculate the mean climatology expected for each SSP scenario and bi-decadal period between 2020–2039 and 2080–2099 at a specific site, we added monthly anomalies for that bi-decadal period to mean monthly temperature and total monthly precipitation during the normal period 1995–2014. Next, to simulate the effects of climatically extreme years on future tree growth we determined the 0.15 quantile of temperature and the 0.75 quantile of precipitation for each calendar month within the normal period 1995–2014 as a surrogate for historical cool-wet years. Similarly, warm-dry years were defined using the 0.85 quantile of temperature and the 0.25 quantile of precipitation. We added monthly SSP anomalies to monthly climatologies of both historical extremes to predict precipitation and temperature during future extremes for each bi-decadal period. This process yielded 48 datasets of future climatic conditions for each site (4 SSP scenarios x 4 bi-decadal periods x 3 climatologies).

## Modeling intra-annual wood formation using the VS-Lite model

VS-Lite is an empirical model capable of simulating intra-annual progress of tree-ring formation with monthly temporal resolution. It assumes that the monthly growth rate of a mean tree per species in a specific stand is driven by temperature, soil moisture, and photoperiod[31]. Accordingly, inputs into the model include monthly mean air temperature, precipitation totals, and site latitude. In the first step, the model simulates monthly relative volumetric soil moisture from temperature and precipitation using the leaky-bucket water balance equations. Next, partial growth rates to temperature and soil moisture are determined using non-linear response functions (Fig. 6a, b). Partial growth rates vary between 0 and 1 indicating no growth due to strong climatic limitation and peak growth under climatically-optimal conditions, respectively. The lower partial growth rate to temperature and soil moisture for each month serves as a proxy for the rate of wood formation building on an assumption that wood formation is sensitive to the most limiting factor (Liebig's law). Accordingly, an integral growth rate, i.e., a unitless proxy of monthly growth rates, is calculated from the lower of the two partial growth rates, representing temperature and soil moisture, multiplied by the partial growth rate to photoperiod (Fig. 6c). The latter depends on the mean monthly daylength calculated from site latitude. The annual tree-ring width is proportional to the sum of integral growth rates over the year which can optionally also account for autocorrelation effects from the previous year (Fig. 6d).

For this research, we modified the original algorithm of VS-Lite according to the recommendations of previous studies. First, we increased the complexity of the response function to permit the negative impacts of extreme heat on radial growth (Fig. 6a)[45]. We also weighted the previous-year effects on the current tree ring to account for autocorrelation in simulated tree-ring width chronologies (Fig. 6d)[34]. A detailed description of both modifications and their implementation is provided in ref. 69. Since the evapotranspiration model simulating soil moisture can have significant impacts on VS-Lite forecasts[33], we tested the original Thornthwaite model with the heat-load index defined based on mean climatic conditions over multiple decades as well as a modified approach calculating heat-load index individually for each year. There were marginal differences between both trials. Accordingly, we calculated the heat-load index separately for each calendar year of the calibration period 1961–2020 and each bi-decadal period for SSP forecasting scenarios.

## Model calibration and validation

We calibrated the optimal parameters of the model for each site over the period from 1961 to the last year of the given chronology (1995–2020). To do this, we randomly generated 10,000 combinations of eight parameters for each site within their ecologically reasonable

intervals (Supplementary Table 3) and, accordingly, produced 10,000 possible simulations. We retained the simulation that resulted in the highest correlation between observed and simulated chronologies at a given site. After the calibration, we ran the model with the calibrated set of parameters and historical climatic data to produce simulated annual and monthly growth rates for the entire 1961–2020 period for each site.

We used two approaches to independently validate the simulated intra-annual growth patterns. First, we employed data recorded by automatic point (TOMST) or band dendrometers (Environmental Measuring Systems) installed at some of our sampling sites in the Czech Republic within the period 2012–2020. We compared simulated integral growth rates with sub-annual records on stem dimensions from 57 sites with 83 independent combinations of site and growing season (Supplementary Fig. 15). To do this, we filtered out the influence of reversible oscillations in stem size due to changing stem water content from dendrometer records using the zero-growth approach[27]. Values of radial growth were averaged from individual trees to the site level, aggregated per month, and standardized by the annual sum of growth rates, i.e., by the total annual growth. For each site, we plotted monthly standardized radial growth recorded by dendrometers and simulated integral growth rates produced by the VS-Lite model to visually compare their agreement. We also calculated Pearson correlation coefficient, root mean squared error, and temporal offset of seasonal growth peaks (i.e., months with highest growth rate within the year) to statistically evaluate coherence between integral growth rates and dendrometer data.

In addition to dendrometer data available only for a limited number of sites and years in the Czech Republic, we compared simulated integral growth rates with the intra-annual pattern of the Normalized Difference Vegetation Index (NDVI) at all sites. The NDVI was previously used as an indirect proxy for leaf phenology at the ecosystem scale, which is often consistent with spring cambial phenology at the stem level when considered at monthly temporal resolution[34]. Time series of NDVI were obtained using Google Earth Engine for each site and each year between 2000 and 2020 from the MODIS satellite (Terra Vegetation Indices; MOD13Q1). This dataset contains global layers of NDVI calculated from atmospherically corrected images (erased clouds, heavy aerosols, and cloud shadows) with a spatial resolution of 250 m and a 16-day temporal resolution. Similar to dendrometer data, we plotted intra-annual patterns in NDVI and simulated integral growth rates averaged per genus to visually compare their coherence. We also identified the first and the last point during the year when NDVI crossed 0.5 of its annual amplitude as a proxy for the onset and cessation of the growing season[34]. We compared their timing with simulated integral growth rates for the spring and autumn months.

### Growth forecasting

In addition to the baseline calibration period 1961–2020, we used the model to predict tree growth for four bi-decadal periods between 2020–2039 and 2080–2099. To do this, all forecasted climatic datasets were used as an input for the calibrated VS-Lite model to simulate future growth during mean, warm-dry, and cool-wet years, within each bi-decadal period, and under four SSP scenarios. To account for autocorrelation effects in tree-ring width chronologies, we ran ten consecutive simulations with the same climatic data and retained the last year of outputs for each prediction.

### Statistical analysis of the baseline and forecast growth simulations

We used simulated integral growth rates as a proxy for monthly radial growth, and partial growth rates to temperature and soil moisture to quantify climate-growth limitation for each month. According to the model assumptions, the lower partial growth rate identifies a climatic factor limiting growth for each month which, in the case of the

modified VS-Lite model, can be low temperature, high temperature, or low soil moisture (Fig. 6c). We aggregated growth limitations by high temperature and low soil moisture into limitations due to drought stress. Moreover, we calculated a 'growth deficit' for each month as the difference between the simulated integral growth rate (i.e., realized growth) and the partial growth rate to the photoperiod (i.e., the growth that would occur under optimal temperature and soil moisture; Fig. 6e). We standardized growth deficits with the annual sum of partial growth rates to the photoperiod to calculate the relative proportions of tree rings that were not formed due to climatic limitations. We also calculated the relative frequency of growth cessation due to low temperature or drought stress for each calendar month. Growth cessation occurred if the integral growth rate equalled zero, i.e., no growth was simulated in a given month. Finally, we estimated the growing season duration for each year from the intra-annual pattern of cumulative integral growth rates (Fig. 6f). The start and cessation of the growing season were assigned to the month when cumulative integral growth rates reached 2.5 % and 97.5 % of their annual cumulative total.

We used scatterplots and generalized additive models to characterize the relationship between annual growth deficits, the timing of the growing season, and the mean climatic water balance of the site. We preferred generalized additive models for this purpose considering their flexibility towards potentially non-linear shifts in intra-annual growth with climatic water balance. We performed this comparison separately for the baseline period 1961–2020 and each bi-decadal period, SSP scenario, and climatology (mean, cool-wet, warm-dry) to assess the effects of future climate change on growth deficits and phenology. In this paper, we report only baseline data from 1961–2020, forecasts based on the mean climatology for 2040–2059 and 2080–2099, and forecasts based on extreme climatologies for 2080–2099 as representative periods highlighting shifts over the full period and expected amplification of climatic extremes toward the end of the 21st century. Next, we plotted mean integral growth rates for each calendar month, genus, and three clusters of sites determined by the mean annual climatic water balance in the baseline period (dry, moderate, and humid sites with -160–+200 mm, 200–700 mm, and 700–1230 mm, respectively) to visualize simulated intra-annual growth patterns. We compared these plots for the baseline period with scenarios of future climate including future climatic extremes to highlight predicted shifts in seasonal growth rates and phenology. Student's t-test was used to compare forecast integral growth rates with integral growth rates for a given calendar month in the baseline period. We also compared numbers of local maxima in integral growth rates (i.e., growth peaks during the year) among genera, SSP scenarios, and between the baseline period and decades toward the end of the century. The local maximum was defined as a month in which the integral growth rate exceeded those of both the preceding and following months (Supplementary Fig. 13). Finally, we divided the simulated tree-ring width for each forecast period, SSP scenario, and climatology by the mean simulated tree-ring width in the baseline period (Fig. 3, Supplementary Fig. 6). We used Student's t-test to assess the significance of forecast changes in simulated tree-ring widths from the baseline mean for each site. For both applications of the Student's t-test (i.e., monthly integral growth rates and annual simulated tree-ring widths), we used the one-sample, two-sided version with heterogenous variance (Welch approximation) to test whether forecast for a given bi-decadal period significantly differs from the average of the baseline period (1961–2020).

### Testing model stationarity

We used the bootstrapped transfer function[70] in an independent trial to test the temporal stationarity of VS-Lite, i.e., the precision of simulated growth trends outside the calibration period. To do this, we split the period from 1961 to the last year of each observed chronology into

two independent halves and calibrated the model only in the first subperiod. Next, we used calibrated parameters to forecast growth into the latter subperiod. For both subperiods, we calculated linear models between observed and simulated chronologies and compared their intercepts, slopes, and coefficients of determination. Bootstrapping was used to quantify the significance of changes in these parameters between subperiods with a special focus on regression slopes because this parameter is linked to the stationarity of growth trends[70]. Note that the temporal span of our chronologies was below the recommended minimum for the application of the bootstrapped transfer function which increases the risk of falsely significant non-stationarities.

### Reporting summary

Further information on research design is available in the Nature Portfolio Reporting Summary linked to this article.

## Data availability

The main inputs and outputs of the wood formation model have been deposited in the Zenodo database under accession code (https://doi.org/10.5281/zenodo.16931899). Main raw tree-ring width series are deposited in the TreeDataClim database (https://treedataclim.cz/en/database/; including sites from the CzechTerra) and RemoteForests database (www.remoteforests.org). The current climatic niche presented in Fig. 1b is based on chorological data available from (https://data.mendeley.com/datasets/hr5h2hcgg4/6). Historical climate from the E-OBS gridded dataset was accessed from (https://surfobs.climate.copernicus.eu/dataaccess/access_eobs.php#datafiles) and climate anomalies from CMIP6 from (https://climateknowledgeportal.worldbank.org/netcdf-browser?prefix=data/cmip6-x0.25/). NDVI data were accessed through (https://lpdaac.usgs.gov/products/mod13q1v061/). Following software was used for data collection and preprocessing: PAST4, TSAP-Win, WinDendro, CooRecorder, Mini32, Lolly Software. Statistical analyses were performed in R 4.2.2. using packages 'dplR' (processing of dendrochronological data), 'dendRolAB' (non-stationarity test), 'mgcv' (generalized additive models), and 'ggplot2' (charts plotting). Source data are provided with this paper.

## Code availability

R-scripts implementing the VS-Lite model have been deposited in the Zenodo database under accession code https://doi.org/10.5281/zenodo.16931899.

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

## Acknowledgements

This work was funded by the Czech Science Foundation [24-11757S], Charles University [PRIMUS/24/SCI/004], and Program Johannes Amos Comenius P JAC [CZ.02.01.01/00/22_008/0004605] awarded to J.T. In addition, M.Ryb. and T.K. acknowledge personal support from the Czech Science Foundation [23-07583S], and M.Š. was supported by the Ministry of Agriculture of the Czech Republic [QL24020351].

## Author contributions

J.T. and V.T. designed the concept of the study. J.T. performed the statistical analysis, interpreted the results of the model, and drafted the manuscript. V.T., J.K., E.C., J.D., J.A., L.P., N.A., M.Ryd., I.V., M.Š. and J.J.C. contributed critically to the drafts and conceived new ideas. All coauthors commented on the manuscript, improved its quality during discussions, and approved its final version. J.T., J.K., J.A., E.C., V.Č., T.Č., J.D., P.F., P.J., R.K., T.K., J.L., J.M., K.N.H., L.P., M.Ryb., M.Ryd., R.S., M.S., M.Š., P.Š., I.V., M.V. and V.T. contributed to the dataset of raw tree-ring width series used in the analysis (including their collection in the field

and laboratory processing). Dendrometer data were contributed by J.K., V.T., M.Š., J.D. and R.M. NDVI data were extracted by J.M.

## Competing interests

The authors declare no competing interests.

## Additional information

[1]Department of Physical Geography and Geoecology, Faculty of Science, Charles University, Albertov 6, Prague, Czech Republic. [2]Department of Forest Ecology, Landscape Research Institute p.r.i., Lidická 25/27, Brno, Czech Republic. [3]Institute of Botany of the Czech Academy of Sciences, Dukelská 135, Třeboň, Czech Republic. [4]Department of Forest Ecology, Czech University of Life Sciences, Kamýcká 129, Prague, Czech Republic. [5]Faculty of Science, University of South Bohemia, České Budějovice, Czech Republic. [6]Instituto Pirenaico de Ecología (IPE-CSIC), Avda. Montañana 1005, Zaragoza, Spain. [7]IFER – Institute of Forest Ecosystem Research Ltd., Cs. armady 655, Jilove u Prahy, Czech Republic. [8]Global Change Research Institute of the Czech Academy of Sciences, Bělidla 986/4a, Brno, Czech Republic. [9]Forestry and Game Management Research Institute, Strnady 136, Jíloviště, Czech Republic. [10]Faculty of Forestry and Wood Technology, Mendel University in Brno, Zemědělská 3, Brno, Czech Republic. [11]Department of Environment, Faculty of Environment, University of Jan Evangelista Purkyně, Pasteurova 15, Ústí nad Labem, Czech Republic. [12]The Green Concept, Institute for Carbon Assessments and Restoration Ecology, 101 Mohar Apartments, Pune, India. ✉e-mail: tumajerj@natur.cuni.cz

