## [Peer Review file · Nature Communications]

Longer growing seasons will not offset growth loss in drought-prone temperate forests of Central-Southeast Europe

Corresponding Author: Dr Jan Tumajer

Version 0:

Reviewer comments:

Reviewer #1

(Remarks to the Author)

Overall, this is an interesting topic in which the authors examine if longer growing seasons in future due to climate warming will impact radial growth in a number of conifer and deciduous species in central Europe. It is well written and all tables and figures are necessary and informative.

However, the novelty of the research is not convincingly portrayed in the manuscript. A quick google search found a relevant paper, by one of the authors, that should have been cited which indicates that a longer growing season does not necessarily result in greater radial growth in pine trees in Boreal and Mediterranean areas (not in temperate climates), Camarero et al 2022, Agricultural and Forest Meteorology <https://doi.org/10.1016/j.agrformet.2022.109223>
The manuscript is interesting but in my opinion not sufficiently novel nor substantive for Nature Communications. However, I do think it could be published in a different journal if the message is clearer, the relevant literature cited, more details provided and a clear indication if differences between genera/time periods/scenarios were statistically significant.

Abstract

L45 indicate if these are deciduous, coniferous,
L49 define which high-emission scenarios are being used – RCPs? SSPs?
What is the time period being examined?

Were the same genera impacted by growth acceleration in mountain areas as those impacted by growth reduction in lowland areas.

Since this manuscript is focused on temperate forests I think it would be useful to mention any risk of not meeting chilling requirements.

Introduction

It would be useful to provide more information and context for the study. The introduction, as presented, is very general. It would be useful to expand on the climate variables that impact growth and development – spring temperature is of course a key variable but winter temperature is also important to ensure chilling requirements have been met. It would also be useful to mention differences in phenology and rate of growth between species/genra with varying wood anatomy type.
Much of the last paragraph in the introduction reads like materials and methods.

Overall, the introduction could do a much better job at presenting the issue more clearly, including presenting species-specific supporting data from the literature and how the current findings will advance the field. What is new and novel about this research? What will the results be used for? A longer growing season may partially compensate for negative impacts of summer drought on growth – is not a new concept.

Materials and methods

I am not sure all the detail about the SSPs is necessary – a summary table would suffice.

Results

Figure 2: what is the time period used

L147-160 were the differences statistically significantly different?

It was not possible to know if the differences between time periods or scenarios were significantly different.

Discussion

L217 is the use of the term 'significantly' quantifiable?

A section on the limitations of the study would be beneficial.

(Remarks on code availability)

Reviewer #2

(Remarks to the Author)

The authors calibrate the VS-lite model to then forecast under a the available climate scenarios the tree ring growth increment of a large number of species of different wood anatomy and phenology. Specifically, they investigate the dynamics that create the ring width index (a measure that indicates a simulation year's anomaly from the baseline period's ring width) through the growth deficit relative to optimal (unlimited) growth, and also through growing changes in season length.

They find that growth-responses are climate change scenario, species and elevation dependent. They find a general trend to less productive low-land and more productive high-land areas, but discuss the latter critically. They point out that these shifts in growth-dynamics may have implications for forest structure and other variables such as evapotranspiration.

My feedback is mostly asking for clarification and further explanation of the methods applied, and the interplay between results and discussion; some more analysis may be necessary. The interpretation of the results seems generally reasonable, but in some cases I am not sure the reader can deduce the results that are discussed from the figures and analysis available. Maybe this can be mitigated by better referencing between figures and text, maybe additional analysis (see below) is needed. As it stands, since I cannot make total sense of the results, I would reject the paper. I am also unsure whether ist angle and scale is currently relevant enough for Nature Communications, as the work is currently more descriptive and therefore specific to the region. I would expect more extrapolation to global scale or generalising of trends (which, granted, has been done to some extent). If more discussion around the mechanisms and model were added, the results could in my opinion be better generalised. The dataset is indeed impressively large, but there could be more efforts made to highlight the low number of broadleaf sites used for drawing conclusions.

Here my comments:

The results have not convinced me that these trends or decadal means of ring width deviations are unprecedented. I see two occasions where ring width deviations will go beyond the observed variability of the baseline period and that is for abies and piecea in the low lands for the two high emission scenarios. Maybe the use of the full set of annual simulations would be more impressive, or a range bar on your decadal mean symbols.

The authors have done well in adressing previous concerns with the model, such as implementing a high temperature-sensitivity and previous-year effects not accounted for in the initial model version. Nevertheless, it would have been useful to include in the discussion some sentences on the original use off he VS-model for conifers and what implications that has for applying and interpreting the results and dynamics angiosperm wood anatomy. Some information may lie in the parameterst hat emerged for angiosperms vs conifers post-calibration, but the study does not go into that detail.

Why was the dendrometer data not used as source of calibration? It is indeed the intra-annual variation of growth that is the strength of applying the VS-lite model (and in my opinion a weakness when calibrating the VS-lite model without intra-annual data). When only calibrating against the final normalised annual ring width, you run into a problem of equifinality (many parameter combinations can lead to your desired outcome of annual width) that can be further constrained when using intra-annual observations. If you can convincingly argue for why this should not be the case (and I think there are reasons, but they are not discussed), then I will drop this criticism, but right now I see this as a fundamental problem in the methods.

The method description would need clarification on a few points in the historical and future climate forcing used, see below comments following L 363ff

There are inconsistencies in how the methods, results and discussion work together around the concept of „growth deficit“: Growth deficit according to the methods is the difference between optimal growth and realised growth. This has implications for interpreting Figure 4 row a and b.

Discussion line 251 describes the growth deficit differently, as growth inactivity.

Currently I don't actually think that the results are presented in such a way that the drought- effect on the reduction of growth in summer can be reproduced by the reader. While I don't doubt that the authors may have found this effects, I have difficulties verifying through your results that the dip in growth causing a „bimodal“ response is indeed driven by summer droughts. Maybe a correlative analysis between summer growth rate and soil water modifier or a simple series of plots with the coinciding of the total growth rate with the VS-lite growth-rate modifiers by temeprature and precipitation could help. Maybe a modification of the local maxima frequency – analysis could also be of use to highlight the effect of precipitation on this „bimodal“ distirbution.

Generally, I find that the analysis and interpretation here would benefit from some more in-depth analysis and presentation of the factors that cause growth onset and cessation. This is not directly extractable from the figure 4 or S7. There we only see

the (seasonal mean ? of the) growth deficit caused by the individual factors and can speculate what the start and end may be caused by.

Line 127 – something missing in the sentence. „observational“ „calibration“ „verification“ sites?

Line 699 and L133ff: I feel I need an explanation why 0.5 is a useful NDVI-threshold value to compare to radial growth onset and termination. Whether by having chosen using optimisation or for physiological reasons or by referencing another study. I cannot find the 0.5 threshold condition in the citation given in line 449 either, and it is unclear why that reference is provided there.

L709. For „a“ given „elevational“ belt.

L 145: decoupling to me is a very mechanistic term. The scenarios and elevations are per default decoupled, they have nothing to do with each other. Maybe „divergence“?

L145 ff: having a problem with the words „ simulated annual growth rate“. If you refer to figure 3, as you did for the previous sentence, then these are not growth rates, but ring widths. And as you rightly say (and one reason for why you apply VS-lite I assume) is that you want to disentangle rate and duration of ring width. Here, I cannot see, using Fig3, why you are explaining the dynamics in Fig 3 ssp2 4-5 with an annual growth rate. Maybe add the correct supplementary material figure or table for this or rephrase to : simulated annual „ring width“ at the end of the century under the low emission SSP2-4.5 scenario „increased“ in high..

L151: but „an“ increase

L167 replace „ will become balanced“ with „ are projected to balance“

Figure 4: add a „ start“ and „end“ into the graph in Figure 4 c) (1), it was not immediately clear that the two lines(groups of data) meant start and end of growing season. Or refine the y-axis title?

L 175 unclear phrasing, which is maybe symptomatic of a problem I have with the whole rest of the paragraph (see long-text about recommended additional analysis) I understand that the extension was driven by an earlier start of cambial activity, but then it reads to me as if something other than cambial activity may drive the end of the growing season. VS-lite is built on assumptions around cell production / cambial activity (ON, OFF, or active with certain temperature+water-dependent rate). The end of the growing season is therefore also driven by the and earlier or later cessation of cambial activity. So the end of the growing season is unambiguously driven by cambial activity in the VS-model. Maybe you mean that it is ambiguous what climatic drivers are responsible for the end? Or do you simply mean that there is a non-linear behaviour with elevation? What would be really interesting- and what is within the scope of your model output –is related to my next comment.

L 179 „Due to amplified drought stress“- has no reference to any figure but is maybe the most interesting part. What – according to the model drives the onset and cessation? In my understanding Figure 4 a and b show the general growth deficits over the full season and we cannot see the drivers of start and stop for c). from Fig4 graph or S7.

I think a more in-depth analysis on what environmental factor is 1) dominating or immediately active after growth start then 2) causes the cessation growth at the end of the season would be a more useful way to interpret figure 4 row c).

L 721 integrate legend on black blue and red line into the figure,maybe in an empty panel

L190

From the figure you refer to here I am not convinced it is fair to say that the conifers look completely bimodal (i.e. similar height peaks with a strong trough between), and I don't think the Broadleaves can be described as completely non-bimodal. This is what I see:

Broadleaves are largely *left*-skewed, in some cases with a shoulder or small secondary peak.

Conifers are more strongly tending towards a bimodal growth pattern.

Additional thoughts:

In the Figure S9, I would agree that overall there are bimodal features, but these are frequency distributions of all sites combined I assume. There, I would even say that Broadleaves also have bimodal features. How can these two results be consolidated?

L217 change „decoupled“ with „diverged“

L220 The results have not convinced me that these trends or decadal means of ring width deviations are unprecedented. I

see two occasions where ring width deviations will go beyond the observed variability of the baseline period and that is for abies and piecea in the low lands for the two high emission scenarios.

L 230 replace „by „ with „in“

L239 ff: „onset“ not quite right, and „decoupling“ not right. An attempt:

Indeed, high-emission SSP scenarios might cause continuously declining lowland forest growth and benefit (?) mountain forests by causing potentially fast turnover.

L 321 „models“ to „model“

L 351: growth deficits are an inactivity? Methods say otherwise. Replace „inactivity“ with „relative reduction from optimal growth rate“?

L 363ff

It is unclear in these methods how the authors have combined the projection climate dataset and the historical climate data. It reads to me that (spatially interpolated) historical time series were obtained from climatological stations. The baseline period for the climate anomalies does not correspond to the baseline climate used for calibration of the VS-lite model. I wonder whether this causes problems with the simulations. It is further unclear whether statistical downscaling was performed on the low-resolution CMIP6 forcing data to accommodate high and low- latitude differences, which normally manifest at high-resolution, as realistically as possible.

L 422 with „the“ calibrated

L 471: replace „by“ with „using“

L 493 add „(Figure S9)“ somewhere.

L 493: unclear where in the results this is reported. I only found a difference from the baseline in relative ring width in Figure 3, but nothing related to phenology and kinetics. Best highlight to the reader which figure you are referring to.

L 496 ff

I cannot find any tables or plots related to this analysis. Can you please better connect it to your results/supplementary materials?

L 703: needs explanation that the SSP symbols are also decadal averages (if they are).

(Remarks on code availability)

Reviewer #3

(Remarks to the Author)

General comments

Tumajer and co-authors presented a manuscript on the radial growth of tree species in Central Europe. The paper is well-written, presenting a timely and interesting topic in the context of forest and climate change. It introduces a novel approach to calibrating a tree-growth model (VS-Lite) using remote sensing and dendrometer data. The innovative use of remote sensing for model calibration is particularly noteworthy given the increasing global availability of such products. The analysis are robust, and complemented with an online code that allows reproducibility. However, details of the methodological process, as currently described, might be difficult to comprehend for readers who are not familiar with such models and their approach.

Although the methods are solid, the suitability of this study for a journal such as Nature Communications needs consideration. The title and discussions suggest an analysis at a global or continental scale, since authors generalize their results for temperate forests. However, the analysis was only performed in sites within Czech Republic. This is certainly still valuable, but the way this is presented to the readers is misleading. No mention to the study region is given until the Material and Methods (line 336). A map of the sites is somewhat hidden among the last figures in the supplementary. Results are therefore not fully representative of Central Europe as a whole, and neither for temperate forests, as the title and abstract suggest. Incorporating data from the Alps and other lowlands in Central Europe could broaden the elevation range. If this is not possible, it should be made clear from the beginning of the manuscript what are the boundaries of the input data, and generalizations of the results must be amended. If that's the case, also the title should reflect the geographical limitation of the study region.

There is also a question of whether a monthly-resolution model is sufficiently detailed, as minimal effects of climate change until 2050 may result from this coarse time resolution. Additionally, there should be a discussion on how relevant growth reactions to average future climate conditions are, given that forest growth appears more influenced by climatic extremes (e.g., sudden drought spells, heat waves, subsequent dry years) rather than chronical temperature increase as indicated by

climate projections. More generally, the paper is missing of a discussion paragraph about the main limitations of the study (e.g., limited representativeness of the sites for Central Europe, intrinsic model limitations, tree mortality not modelled, climate extremes).

Furthermore, I have some doubts about the robustness of the dendrometer data used for calibration. The calibration sites (57 sites) are not shown and little details about them are given in the manuscript. Therefore their representativeness for the environmental gradient cannot be assessed. Additionally, most dendrometer data only cover one year, which may not represent the historical climate period. This should be clarified by the authors.

I provide here below a list of more specific comments that can help to improve the manuscript.

Specific comments

L.46 "future CO2 emissions". Although this is correct, it sound like the model uses CO2 time series, but this is not the case. Perhaps write just "future climate change".

L. 50-53 "mountain forests" and "lowland genera". The title of the paper suggests a focus on lowland forest but the abstract describe differences between mountain and lowlands. This is confusing. It is also not clear at this stage what the author mean by lowland genera and how they are differentiated.

L. 56 "sustainable". Not self-explanatory. Not really clear at this stage what does it mean "sustainable" in terms of stem growth. Please clarify.

L. 84 "negative trends" I am unsure if "trends" is the correct term to be used here. Do you mean negligible increase in total growth even if timing of growth onset is anticipated? Please clarify.

L.96 "instant" What do you mean by "instant"?

L. 96 Figure 1. Overall, I like this comprehensive figures about the functionality of VS-lite but it might be difficult for readers who are not familiar with the model to understand all parameters and functions. First of all, you referred to a modified version of the model, and at this stage the model is not described yet (i.e., what has been modified? Modifications were not described yet). Furthermore, when at the end of the caption you describe the parameters calibrated at each site, sites and study design it has not been described yet. This is confusing. Additionally, in Fig.S8 it is written that parameters explanation is reported in Figure 1. Parameters are displayed, but not explained. For readers who are not familiar with the model, this is hard to understand. Please clarify.

L. 105 "drier lowlands to the alpine treelines". When I read this I expect this study to cover a gradient from the Mediterranean to the upper treelines in the Alps. However when I look at figure 3 I realize that the maximum elevation range is 1490 meters. I am not sure if this can be considered the upper tree line in continental Europe. Please provide the elevation range and a description of the study sites/region.

L. 106-107 "83 intra-annual series of radial growth". Also this is misleading. By reading this I expect the authors to have used 83 sides with longer time series observation reflecting mean inta-annual growth of the species as a representative for an historical period. Instead, only 56 sites were used and mostly with only one year of observation. Please rephrase it clearly. This also refers to my general concern reported above about dendrometer data.

L. 117 Besides a positive effect on radial growth, an earlier start of the growing season could also lead to a higher vulnerability to late frosts with low growth rates in years with frost damages. I believe this is not considered VS-lite due to its monthly resolution. Please acknowledge this aspect (somewhere in the manuscript).

L. 117-121 "systematic variability" Not really clear to me what it is meant by systematic variability, whether it is referred to only one variable at a time or if it includes all the variables mentioned subsequently. Please provide a more precise explanation of this term and indicate whether it applies to each variable individually or collectively.

L. 125-126 "observed and simulated tree ring width chronologies" For the calibration period 1961-2020?

L. 127 Figure 2. When I look at these results, so far I still have no idea about the geographical area of reference. I can only assume that the authors used chronologies from all around the world. This referred to my main concern about the authors not being very clear about the geographical constraints of their application. Being more transparent about the geographic limits of the study would help the interpretation of this figure. Other suggestion: consider adding a legend for the color lines in the figure and not only in the caption description. Quicker to interpret.

L. 129-131 there is also higher peak of growth rate for all species except *Abies alba* in dendrometer compared to model data (which is then offset by higher values for model data in autumn). Having just one year of dendrometer data for most sites (according to Fig. S2) could have quite a big influence on the comparison of dendrometer and model data. Please refer to my general concern mentioned above.

L. 137. "diffuse stands or below-canopy genus" Unclear. I would not generally agree with the characterization of *Abies* sp. (I guess mainly *A. alba*) as below-canopy species, but this probably depends on the specific forest types and stand structure. Perhaps rephrase it into "shade intolerant" and "shade tolerant".

L. 140 Figure S5. I believe that caption in figure S5 would be the right place to fully describe the meaning of the temperature and soil moisture parameter. Please also explain what the other two parameters are. I believe Acor means autocorrelation but this is not clearly explained.

L.142 Figure 3. Upon examining figure 3, it is evident that the elevation gradient of the sites is not representative of the alpine region. Instead, it only reflects central Europe from lowlands to mountain elevation belts. The primary conifer species covering subalpine to tree line elevational zones appear to be absent, such as the Swiss stone pine. Specific comments: 1) what does the color line indicate. If colors are not really used (species are already differentiated by panels) wouldn't it be more efficient to use color lines to differentiate elevation belts and optimize the space? Some boxes are blank (not ideal). By optimizing the figure in this way, size of the panels could be increased enhancing interpretability (i.e., printed on half-page, symbols can be hardly differentiated). 2) I suggest adding an ab-line or a shadowed area in gray to differentiate calibration vs forecasted period; 3) shadowed area within the graph and the color used for the species is barely visible, please enhance.

L. 143-145. I do not understand very well this sentence. Please rephrase.

L. 153 "neglectable" Do you mean "negligible"?

L. 147-149 and L.162 Confusion when referring to low, middle, and high elevations. Please provide numbers about the

elevational gradients and correct (throughout the manuscript) the term “treeline”, which cannot be used as higher site is at 1490 m a.s.l. (unless in Czechia the natural upper treeline occurs at these elevations, but I doubt this is the case).

L. 170 Figure S7. The figure is really hard to interpret. Panels are very small. Consider splitting into two separate figures. Additionally, the fourth panel for scenario SSP5-8.5 seem a repetition of what already shown in Figure 4.

L. 172 Figure 4. Please explain better how to read the phenology (c) graph. It took me a while to understand why there are two trends lines in the panels because it is not clear which line represents the onset and offset of the growing season. Please provide a clearer description in the caption.

L. 177. What do you mean by ambiguous trend? Please clarify, perhaps describing this more directly in the caption. Also, trends are difficult to detect from figure S7 (see two comments above).

L. 182 Figure 5 and Figure S8. Figure S8 is suitable for supplementary material, but Figure 5 should be enhanced to better illustrate the differences between climate change scenarios. For instance, a reduced number of examples can be presented (as the elevation differences are often negligible), and varying line colors or types could be employed to emphasize the distinctions between climate change scenarios within the same panel. In Figure S8 there are panels that repeat results from Figure 5 (necessary?).

L. 195 “cambial activity in the region”. Once more, I have no idea about the region (whole Central Europe?). I doubt whether Materials and Methods were simply shifted to the end of the manuscript after having initially structured the manuscript in the classic IMRaD format.

L. 209-210. This text and Figure S.11 indicates that temperature and precipitation differences are minimal between the four scenarios in the 2050s. Interestingly, in the 2090s, differences in precipitation between scenarios appear greater than those in temperature. However, as the figure shows yearly courses, this might be misleading. A figure showing deviations from the 1961-2020 means (standardized by standard deviations) could help estimate these differences. Additionally, a line plot of temperature trends from 1960 to 2100 would clarify developments. Furthermore, Figure S11 could be enhanced by incorporating annual time series or trendlines to more effectively illustrate the rising temperature trends and variations in precipitation over time, together with panels with intra-annual pattern, without necessarily repeating the baseline for two different periods as it is currently done (confusing).

L. 215-216 “if climatic extremes, including drought spells, (...)”. The recent past especially in Central Europe showed that tree growth and forest vitality is mainly determined by climatic extremes. This limitation should be discussed more extensively (see generic comments).

L. 225 “populations” In forest genetics, populations usually denotes specific adaptations and genotypes. I do not think that is what you mean here. Please clarify.

L. 228-229 “long term growth reductions often precede forest die off”. Agree. This study can only speculate on that issue because the model does not simulate tree mortality. This aspect should be discussed more clearly in the manuscript.

L. 259-260 “indirect climatic effects” unclear what is meant here by indirect climatic effects. Do you intend, for example, natural disturbances?

L. 274-284. I did not find this paragraph very informative, at least in the way it is phrased now. The main question appears to be: will central European species known adapted to this pattern be able to tolerate these repeated reduction or will this lead to tree mortality? I suggest expanding a bit this aspect which is very interesting but in relation with recent studies from literature on tree mortality.

L. 289-291. I am a bit puzzled about this sentence. To my knowledge, the distinction between species with isohydric and anisohydric strategies is not equal between broad leaves and conifers. There are some conifers that are more drought resistant than broad leaves. Please clarify, expand or rephrase.

L. 331-333 “mechanistic understanding or ecosystem responses to future growth dynamics is needed” agree, however this has been addressed in many studies using process based forest models. Please expand a bit more this discussion in the context of recent literature, in particular giving some examples of applications with process based models on forest growth and tree mortality.

L. 348-352. To what extend to you remove long-term growth-trends which might be a reaction to climate change in the past decade by using a spline detrending compared to a more conservative detrending or using BAI? Please clarify.

L. 364 “totals” What does “totals” mean?

L. 371-383 This section describes the IPCC climate change scenarios, but lacks details on how GCM projections were downscaled to the site level. Additionally, it mentions climate anomalies by decade, though the VS-lite model requires monthly time series. More information on the processing and preparation of climate data is needed.

L. 381-383. “we refer to SSP1-2.6 (...)” This sentence is essential for interpreting figures and results but it is mentioned only at this late stage in the manuscript. Once more, it makes me doubting that the manuscript was initially structured in IMRaD format without a final check for consistency. understanding of the results. Please place it earlier in the text.

L. 387-388. Recent studies based on dendrometer data suggest a strong influence of VPD on radial increment/growth. Should this also be accounted for? Sr is the monthly resolution of the model to coarse for it? Please discuss these aspects. Also, there is an “a” is missing between “in” and “specific” in line 387.

L 406-407 The reference should probably be to Fig. 1a or 1c instead of 1b.

L. 429 “83 growing seasons at 57 sites”. Only at this stage of the manuscript it is clear to me that calibration sites with dendrometers data are 57. However, it is still unknown whether these sites cover the environmental range of variability across the study region. I would suggest to move the map, currently in the supplementary, in the main manuscript by adding also the location of the these sites.

Table S1. What are species “Q.sp.” in Quercus and “F. sp.” In Fraxinus? If species is unknown or hybrid/exotic, why was it not excluded from the analysis?

(Remarks on code availability)

I only gave a check to the scripts and code and it appears all on order and reproducible. Upon publication, I suggest archiving the code in a long-term official repository (e.g., Zenodo, Dryad).

Reviewer #4

(Remarks to the Author)

(Remarks on code availability)

I reviewed the R data repository and it appears that all the necessary inputs to run the code are provided. However, I didn't run the code or examine the inputs closely due to time constraints.

Version 1:

Reviewer comments:

Reviewer #3

(Remarks to the Author)

I truly appreciate the efforts by the authors to intensely revise the manuscript. All comments and recommendations were thoroughly addressed and I believe that the manuscript has been greatly improved. I particularly appreciate the honesty in acknowledging the geographical constraints of the study by providing a clear distribution map of the dataset and by revising accordingly the title. Furthermore, extending the dataset to enhance its geographical representativeness and expanding the simulation analysis to include meteorological extremes provided additional emphasis to the results.

I have checked each response made to each of my comments and I am satisfied the new version. I only have a few points that need to be addressed, among them how the two alternative climatologies were derived (cool-wet, warm-dry). This is explained only from line 569 (section Growth forecasting). The two alternative climatologies were derived by assuming a reduction in temperature and precipitation from climate projections. Please move this to the "Climatic data and climatic scenarios" section and refer to it earlier in the manuscript. This adjustment is important as these assumptions affect the simulated outcomes and their interpretation.

I provide some specific comments to the revised version here below:

L.124-125 Please add a brief motivation for adding these two alternative scenarios. Otherwise comes as a surprise for the reader. You could also add "(see details on how scenarios were derived in the Methods section)".

L.151 "diffuse stands" I suggest using the term "sparse". Already commented in previous version, diffuse is ambiguous.

L.155 I think you could simply title is "Growth forecast under climate change projections" as these are the standard projections provided by climate models. Then in the text below you clearly acknowledge that in the manuscript you referred to it as "mean climatology"

L.223 "Models suggested (...)" If you referred to fitted models to the results, I suggest simply writing "Results suggested (...)", to not confuse with model simulations.

L.421-427 Referring to my previous comment ("Please expand a bit more this discussion about process based forests models"). I did not mean to discuss the need to integrate wood formation into stand models. Models can always be made more complex, but integrating fine-grain processes such as wood formation – and therefore more and more parameters - into e.g., stand or landscape forest model, this does not always guarantee that the models would work better for modelling processes. I meant to acknowledge more clearly the limitations of models like VS-Lite in the context of a broader literature on forest process-based modelling. Using a model that includes stress-induced mortality due to climate and disturbances on the same sites, will likely provide a much more dramatic picture than what currently presented in this study. Please add a short paragraph at the end of the Model and database limitations acknowledging these aspects correlated with recent literature (e.g., see Bugmann et al 2019 Ecosphere).

L.475 "we forecast" You did not forecast but "retrieved climate change projections from xx-yy database"

L.480 "Climate Change Knowledge Portal" Please add a reference or link to the resource.

L.492 I suggest adding here the description on how extreme climatologies were derived.

(Remarks on code availability)

Reviewer #4

(Remarks to the Author)

(Remarks on code availability)

Reviewer #5

(Remarks to the Author)

Manuscript: Revised manuscript entitled "Longer growing seasons will not offset growth loss in drought-prone temperate forests of Central-Southeast Europe," by Tumajer et al.

Task: The editor asked me to evaluate the authors' responses to the comments from the initial reviewers. I was not asked to provide an independent review of the revised manuscript. I was also asked to specifically focus on the response to comments from Reviewer #1 (R1) and Reviewer #2 (R2), since neither was available to assess the authors' responses to their comments. To complete this task, I read the revised manuscript and the authors' responses to the reviewers. I focused my evaluation on the major issues.

Summary: Overall, the revised manuscript addressed the major concerns of the reviewers. To do so, the authors expanded their focus with additional data and conducted additional analyses. However, the manuscript would still benefit from a more thorough explanation of its statistical framework and more design attention to figures.

Evaluation of major concerns:

1. Major shared concern: Not appropriate for the journal

R1 thought the paper lacked sufficient novelty

Response: The authors added new components that expanded prior work including new analyses exploring potential tree growth responses to future climatic extremes and expanded their literature the addressed the importance of understanding differences in response among species.

R2 thought the paper had too much of a regional focus.

Response: Authors expanded their dataset to new sites that expanded the spatial inference to sites in Eastern Europe and the Balkans and included more sites with broadleaved tree species.

Evaluation: Authors made a good faith effort to address the concerns.

2. Major shared concern: Difficulty in making sense of the results

R1 noted several specific instances where they had trouble following the methods and results.

R2 noted in their general comments that the results were hard to interpret in part because of a lack of clarity in the methods.

Evaluation: I read the revision de novo. With some careful reading, I could track how the results were obtained from the methods. However, it is challenging to follow the results without first reading the full Method and Material section. Also, the graphics are dense and not always sufficiently explained. For example, Fig 1b – climate variables on axes not entirely explained. Not sure why both r and R2 were presented in Fig. 2 – seems redundant. And splitting out the fits for the different datasets also seemed like an unnecessary detail.

3. R1 identified concerns about the meaning of significant differences.

Evaluation: The statistical analyses used in this paper are not fully described and the responses to the R1 comments did not clarify the situation. I am not saying the stats were inappropriate, but the choice of metrics (t-tests, GAMs) is not justified nor is sufficient analytical details provided.

4. R2 had questions about model calibration and validation. The main concern was using annual ring widths rather than monthly growth to calibrate the VS-lite model.

Evaluation: Authors noted that intra-annual data from dendrometers was ideal but such data was not available at the scale of the study. Their alternative – calibrate with annual and validate with intra-annual – struck me as a reasonable compromise. That said, a more thorough evaluation of the validation metrics (Fig. 2b) is required. How good is good enough for the questions they are asking? There is a timing difference and maximum growth difference.

5. R2 expressed concerns about the performance of the VS-lite model and its ability to predict growth of hardwood species.

Evaluation: Revision included the details and caveats requested.

6. R2 wanted more details on how drought-effects during the summer were quantified. This question was related to the distinction between growth onset and cessation.

Evaluation: Revision included a new analysis and results (Fig. S10) that addressed these concerns.

(Remarks on code availability)

Version 2:

Reviewer comments:

Reviewer #5

(Remarks to the Author)

The revisions regarding my specific concerns were addressed in this revision.

The authors' responses make this manuscript a strong, excellent piece of scholarship.

(Remarks on code availability)

Reviewer #1 (Remarks to the Author):

Overall, this is an interesting topic in which the authors examine if longer growing seasons in future due to climate warming will impact radial growth in a number of conifer and deciduous species in central Europe. It is well written and all tables and figures are necessary and informative.

However, the novelty of the research is not convincingly portrayed in the manuscript. A quick google search found a relevant paper, by one of the authors, that should have been cited which indicates that a longer growing season does not necessarily result in greater radial growth in pine trees in Boreal and Mediterranean areas (not in temperate climates), Camarero et al 2022, *Agricultural and Forest Meteorology* <https://doi.org/10.1016/j.agrformet.2022.109223>

Response: Thank you for your valuable comments on our manuscript. We carefully revised the entire manuscript to streamline our research question and stress its novelty. For instance, the second paragraph of the *Introduction* was extended considering the new focus of the study on future climatic extremes and their impacts on growth (L. 79-95). We included a literature review focused on between-species differences in growth phenology and how they might translate into recent trends in tree-ring widths, mechanisms of non-linear effects of climate warming on spring phenology (chilling-forcing interaction, late-frost risks), and dependence of impacts of drought spells on the phase of growing season with their occurrence. Furthermore, we included the reference suggested in your comment within this part of the *Introduction*.

The manuscript is interesting but in my opinion not sufficiently novel nor substantive for *Nature Communications*. However, I do think it could be published in a different journal if the message is clearer, the relevant literature cited, more details provided and a clear indication if differences between genera/time periods/scenarios were statistically significant.

Response: As already mentioned above, we reformulated *Introduction* to better highlight the importance and novelty of our study. We also expanded the spatial domain of our dataset toward Eastern and Southeastern Europe to increase the regional representativeness of our results (Fig. 1). The extended dataset representatively covers main species in the region across a substantial climatic gradient from dry lowlands on rocky soils of the southern Balkan Peninsula to cold Carpathian treelines. Next, we strengthened the novelty aspects by incorporating simulations of intra-annual growth dynamics under climatic extremes, whose intensity and frequency is expected to increase toward the end of the 21st century. Understanding trees' ability to cope with future climatic extremes is essential since historical drought spells were shown capable of initiating forest die-off in dry regions. Additionally, we used appropriate statistical tests to evaluate statistical significance in forecast trends in tree-ring widths and shifts in intra-annual growth patterns (L. 618-619, L. 626-627, Figs. S7, S11).

Abstract

L45 indicate if these are deciduous, coniferous,

Response: We reformulated the statement accordingly. Now we state that our dataset comprises ten broadleaved and five coniferous genera in the *Abstract* (L. 45).

L49 define which high-emission scenarios are being used – RCPs? SSPs?

Response: We revised the sentence accordingly. We specified the use of SSP scenarios in the *Abstract* (L. 46).

What is the time period being examined?

Response: We indicated the forecast period in the *Abstract*. The respective sentence reads: *We calibrated the VS-Lite growth model using 2,013 tree-ring chronologies from ten broad-leaved and five coniferous genera in Central-Southeast Europe to predict intra-annual wood formation under four SSP climate scenarios through the 21st century.* (L. 44-47).

Were the same genera impacted by growth acceleration in mountain areas as those impacted by growth reduction in lowland areas.

Response: We specified that models forecast growth acceleration of coniferous genera in humid sites (L. 49-51), while both conifers and broadleaves are forecast to reduce growth in dry sites (L. 51-53).

Since this manuscript is focused on temperate forests I think it would be useful to mention any risk of not meeting chilling requirements.

Response: We included the non-linear effects of increasing temperatures on spring growth phenology, focusing mainly on chilling-forcing interaction and increasing risks of late-frosts, in the literature review in the *Introduction* (L. 84-87) and also in a novel paragraph of *Discussion* on limitations of the VS-Lite (L. 402-406).

Introduction

It would be useful to provide more information and context for the study. The introduction, as presented, is very general. It would be useful to expand on the climate variables that impact growth and development – spring temperature is of course a key variable but winter temperature is also important to ensure chilling requirements have been met. It would also be useful to mention differences in phenology and rate of growth between species/genra with varying wood anatomy type.

Response: The *Introduction* was substantially revised according to your suggestions. Most importantly, we extended the second paragraph (L. 79-95) to outline the complexity of intra-annual growth patterns of temperate forests including their variation among conifers, ring-porous broadleaves, and diffuse-porous broadleaves (mainly considering the early onset of growing season in ring-porous broadleaves). We also discussed concerns about the increasing frequency of late frosts with ongoing spring warming, and how the effects of increasing temperature on growth phenology might be non-linear due to declining chilling accumulation. According to the novel focus of our study on future climatic extremes, we discuss risks associated with future meteorological extremes and how their timing interacts with growth phenology.

Much of the last paragraph in the introduction reads like materials and methods.

Response: We preferred to introduce the basic methodological concept of our study here to help readers understand the following *Results* section. Nevertheless, we followed your recommendation and partly shortened and condensed this text.

Overall, the introduction could do a much better job at presenting the issue more clearly, including presenting species-specific supporting data from the literature and how the current findings will advance the field. What is new and novel about this research? What will the results be used for? A longer growing season may partially compensate for negative impacts of summer drought on growth – is not a new concept.

Response: As already outlined above, we reviewed the *Introduction* and substantially modified the second paragraph (L. 79-95), stressing (a) differences in intra-annual growth dynamics between species with different wood anatomy, (b) the possibility of non-linear shifts of spring phenology in a response to spring warming due to lack of chilling, and (c) risks associated with future climatic extremes and how their impacts relate to growth phenology.

Materials and methods

I am not sure all the detail about the SSPs is necessary – a summary table would suffice.

Response: We moved the details to **Table S2**.

Results

Figure 2: what is the time period used

Response: The figure caption was extended to provide the temporal period of each calibration/validation exercise.

L147-160 were the differences statistically significantly different? It was not possible to know if the differences between time periods or scenarios were significantly different.

Response: We implemented t-tests to evaluate differences between baseline tree-ring widths and forecast chronologies toward the end of the 21st century (**Fig. S7**). Similarly, we used t-tests to assess the significance of changes in monthly integral growth rates from the baseline period (**Fig. S12**). Statistically significant ($p < 0.05$) results prevailed. We present outcomes of statistical tests in respective parts of the *Results* section (e.g., L. 174-177, 198-200).

Discussion

L217 is the use of the term ‘significantly’ quantifiable?

Response: We used ANOVA and Tukey's post-hoc tests to assess the statistical significance of differences in predicted growth rates between different SSPs. However, given the large sample size ($n = 2,013$, with $n > 100$ for most combinations of genus-climatic gradient), both tests yielded highly statistically significant differences for the vast majority of tested pairs, i.e., very low p-values. Given the limited usage of p-value for analyzing large samples, we preferred not to show the results of these statistical tests in the manuscript. Instead, we carefully revised our previous statements regarding ‘significant differences between SSPs’ to avoid implying that statistical tests were performed.

A section on the limitations of the study would be beneficial.

Response: We included a new sub-chapter of the *Discussion* stressing the main limitations of the study, mainly concerning (i) representativeness of the tree-ring dataset within the full climatic niche of individual genera, (ii) VS-Lite performance, and (iii) limitations due to monthly resolution of forecasts (L. 361-409).

Reviewer #2 (Remarks to the Author):

The authors calibrate the VS-lite model to then forecast under a the available climate scenarios the tree ring growth increment of a large number of species of different wood anatomy and phenology. Specifically, they investigate the dynamics that create the ring width index (a measure that indicates a simulation year's anomaly from the baseline period's ring width) through the growth deficit relative to optimal (unlimited) growth, and also through growing changes in season length. They find that growth-responses are climate change scenario, species and elevation dependent. They find a general trend to less productive low-land and more productive high-land areas, but discuss the latter critically. They point out that these shifts in growth-dynamics may have implications for forest structure and other variables such as evapotranspiration.

Response: Thank you for your comments that we carefully considered and that helped us to improve the manuscript. All our responses to your comments are detailed below.

My feedback is mostly asking for clarification and further explanation of the methods applied, and the interplay between results and discussion; some more analysis may be necessary. The interpretation of the results seems generally reasonable, but in some cases I am not sure the reader can deduce the results that are discussed from the figures and analysis available. Maybe this can be mitigated by better referencing between figures and text, maybe additional analysis (see below) is needed. As it stands, since I cannot make total sense of the results, I would reject the paper. I am also unsure whether ist angle and scale is currently relevant enough for Nature Communications, as the work is currently more descriptive and therefore specific to the region. I would expect more extrapolation to global scale or generalising of trends (which, granted, has been done to some extent). If more discussion around the mechanisms and model were added, the results could in my opinion be better generalised. The dataset is indeed impressively large, but there could be more efforts made to highlight the low number of broadleaf sites used for drawing conclusions.

Response: We extended our dataset by incorporating new sites from Eastern Europe and the Balkans, which also significantly increased the number of broadleaved sites (**Table S1**). Currently, our dataset consists of 10 broadleaved genera with the highest replication for *Fagus* sp. (n=458 sites), *Quercus* sp. (n=106 sites), and *Acer* sp. (n=44 sites). Including new sites also improved the representativeness of our dataset within species climatic niches in Europe (**Fig. 1b**). Another novel aspect of the revised manuscript is simulations of annual and intra-annual growth patterns under forecast climatic extremes, mainly dry spells and exceptionally cool years (**L. 205-229**). Understanding tree ability to cope with future climatic extremes is essential since historical drought spells were shown to initiate forest dieback in dry regions.

Here my comments:

The results have not convinced me that these trends or decadal means of ring width deviations are unprecedented. I see two occasions where ring width deviations will go beyond the observed variability of the baseline period and that is for abies and piecea in the low lands

for the two high emission scenarios. Maybe the use of the full set of annual simulations would be more impressive, or a range bar on your decadal mean symbols.

Response: Thank you for this stimulating comment. We agree that simulations in a forecasting mode mostly stayed within the historical range observed during the calibration period. However, this was because forecast simulations represented mean growth rates within a given bi-decadal period, but baseline simulations reflected high-frequency inter-annual variation including climatic extremes. To expand our analysis, we predicted tree-ring widths and intra-annual growth patterns for two types of climatically extreme years (cool-wet and warm-dry) towards the end of the 21st century. Our results show that growth reductions caused by climatic extremes might go far beyond historical variability, mainly considering dry-warm years at dry sites and high-emission SSP scenarios (**Figs. 3, 5**). Notably, although for cool-wet years at dry sites, a higher summer growth rate was forecast compared with mean climatic conditions, both cool-wet and dry-warm years will significantly limit growth compared to mean climate at humid coniferous sites by the end of the 21st century. We also report notable shifts toward very early growth cessation during dry-warm years at dry sites with high-emission scenarios of climate change (**Fig. 4**).

The authors have done well in addressing previous concerns with the model, such as implementing a high temperature-sensitivity and previous-year effects not accounted for in the initial model version. Nevertheless, it would have been useful to include in the discussion some sentences on the original use of the VS-model for conifers and what implications that has for applying and interpreting the results and dynamics of angiosperm wood anatomy. Some information may lie in the parameters that emerged for angiosperms vs conifers post-calibration, but the study does not go into that detail.

Response: We included a novel sub-chapter focusing on the limitations of our approach including the limitation of the VS-Lite model (**L. 361-409**). Specifically, we focus on limitations associated with (i) lack of chilling-forcing mechanism for simulating cambial phenology, (ii) monthly temporal resolution of the model, and (iii) slightly better model performance for conifers compared to diffuse-porous broadleaves. The respective sentence reads: *Correlations between observed and simulated chronologies peaked for conifers and Quercus sp. but were the lowest for diffuse-porous broadleaves like Fagus sp. This might reflect the spatial distribution of individual sites in our study region, where conifers and Quercus sp. often occupy climatic margins of forest distribution, including treelines and dry lowlands, while Fagus sp. forms forests in medium elevations. Moreover, better performance for coniferous sites might be a legacy of the VS-Lite and Vaganov-Shashkin models originally designed for simulating the growth of boreal conifers.* (**L. 378-384**)

Why was the dendrometer data not used as source of calibration? It is indeed the intra-annual variation of growth that is the strength of applying the VS-lite model (and in my opinion a weakness when calibrating the VS-lite model without intra-annual data). When only calibrating against the final normalised annual ring width, you run into a problem of equifinality (many parameter combinations can lead to your desired outcome of annual width) that can be further constrained when using intra-annual observations. If you can convincingly argue for why this should not be the case (and I think there are reasons, but they are not discussed), then I will drop this criticism, but right now I see this as a fundamental problem in the methods.

Response: We agree that calibrating the model using annual data and interpreting its intra-annual outputs can lead to biased results due to equifinality. Incorporating empirical observations of growth in both intra-annual and annual scales at the calibration stage would be the optimal solution. However, replication of dendrometer data was low with only 53 of 2,013 of our sites with available dendrometer records, mostly for a single year (**Fig. S15**). Accordingly, we used dendrometer data for independent validation instead of calibration to avoid different calibration approaches at sites with and without dendrometer data. Using scarce dendrometer or xylogenesis data to validate rather than calibrate the model is a common approach in previous studies using the Vaganov-Shashkin model (Buttò et al. 2020: *Frontiers in PS*, Jevšenak et al. 2021: *IJBM*) including our previous trials (Tumajer et al. 2021: *Frontiers in PS*).

The method description would need clarification on a few points in the historical and future climate forcing used, see below comments following L 363ff

Response: See our full response on this matter below.

There are inconsistencies in how the methods, results and discussion work together around the concept of „growth deficit“: Growth deficit according to the methods is the difference between optimal growth and realised growth. This has implications for interpreting Figure 4 row a and b. Discussion line 251 describes the growth deficit differently, as growth inactivity.

Response: We checked the entire manuscript for consistent use of growth deficit, i.e., as the difference between monthly growth rate under optimal conditions (determined by partial growth rate to photoperiod) and simulated integral growth rate. Moreover, to reflect your further comments, we also determined the frequency of growth cessation for each calendar month (i.e., the relative frequency of zero integral growth rates) to complement the interpretation of increasing drought as a driver of increased growth bimodality and earlier end of growing season forecast at dry sites (**Fig. S10, L. 189-192**). We adjusted **Fig. 6e** accordingly to clarify the definition of “growth deficit” and “growth cessation”.

Currently I don't actually think that the results are presented in such a way that the drought-effect on the reduction of growth in summer can be reproduced by the reader. While I don't doubt that the authors may have found this effects, I have difficulties verifying through your results that the dip in growth causing a „bimodal“ response is indeed driven by summer droughts. Maybe a correlative analysis between summer growth rate and soil water modifier or a simple series of plots with the coinciding of the total growth rate with the VS-lite growth-rate modifiers by temperature and precipitation could help. Maybe a modification of the local maxima frequency – analysis could also be of use to highlight the effect of precipitation on this „bimodal“ distribution.

Response: See the next response.

Generally, I find that the analysis and interpretation here would benefit from some more in-depth analysis and presentation of the factors that cause growth onset and cessation. This is not directly extractable from the figure 4 or S7. There we only see the (seasonal mean ? of the) growth deficit caused by the individual factors and can speculate what the start and end may be caused by.

Response: Thank you for raising this interesting point. We calculated the frequency of growth cessation for each calendar month, i.e., the proportion of years with zero integral growth rates within a given month. The results highlight seasonality and prevailing driver of growth cessation and how they shift from dry to humid sites, and with climate change (**Fig. S10**). Most importantly, we illustrate the declining frequency of low-temperature-driven growth cessation in winter but increasing drought-driven growth cessation in late summer and autumn, mainly at dry sites and high-emission scenarios. We present and discuss these new results at **L. 189-192, 303-305**.

Line 127 – something missing in the sentence. „observational“ „calibration“ „verification“ sites?

Response: The sentence was reformulated, now it reads: *During the 1961–2020 calibration period, the mean (\pm standard error) correlation coefficient between observed and simulated tree-ring width chronologies was 0.42 (\pm 0.003). A significant correlation was found at 84 % ($p < 0.05$) and 66 % ($p < 0.01$) of 2,013 sites for which we calibrated the VS-Lite model (**L. 137-140**)*

Line 699 and l.133ff: I feel I need an explanation why 0.5 is a useful NDVI-threshold value to compare to radial growth onset and termination. Whether by having chosen using optimisation or for physiological reasons or by referencing another study. I cannot find the 0.5 threshold condition in the citation given in line 449 either, and it is unclear why that reference is provided there.

Response: We corrected the reference to Seftigen et al. (2018): GEB (**L. 559-561**), where authors used a 0.5 threshold of standardized NDVI to approximate the start and end of the phenological growing season. They also justify this threshold value with “The increase in greenness has previously been shown to be most rapid at this threshold”. Our study employs the same approach for NDVI standardization as used by Seftigen et al. (2018).

L709. For „a“ given „elevational“ belt.

Response: Revised accordingly (**L. 852**)

L 145: decoupling to me is a very mechanistic term. The scenarios and elevations are per default decoupled, they have nothing to do with each other. Maybe „divergence“?

Response: Reformulated to *divergence* (**L. 159**)

L145 ff: having a problem with the words „ simulated annual growth rate“. If you refer to figure 3, as you did for the previous sentence, then these are not growth rates, but ring widths. And as you rightly say (and one reason for why you apply VS-lite I assume) is that you want to disentangle rate and duration of ring width. Here, I cannot see, using Fig3, why you are explaining the dynamics in Fig 3 ssp2 4-5 with an annual growth rate. Maybe add the correct supplementary material figure or table for this or rephrase to: simulated annual „ring width“ at the end of the century under the low emission SSP2-4.5 scenario „increased“ in high..

Response: Reformulated according to your suggestion. Now the sentence reads: *The simulated annual ring width at the end of the century under the low emission SSP2-4.5 scenario increased at humid (35 % increase compared to the baseline period) and moderate sites (12 %), but an increase in mean tree-ring width was negligible at dry sites (4 %).* (L. 165-168)

L151: but „an“ increase

Response: Revised accordingly (L. 167)

L167 replace „ will become balanced“ with „ are projected to balance“

Response: This sentence was dropped to streamline the flow of the *Results* section,

Figure 4: add a „ start“ and „end“ into the graph in Figure 4 c) (1), it was not immediately clear that the two lines(groups of data) meet start and end of growing season. Or refine the y-axis title?

Response: Figure 4 was improved according to your suggestions (two separate rows of the panels for growing season start and end)

L 175 unclear phrasing, which is maybe symptomatic of a problem I have with the whole rest of the paragraph (see long- text about recommended additional analysis) I understand that the extension was driven by an earlier start of cambial activity, but then it reads to me as if something other than cambial activity may drive the end of the growing season. VS-lite is built on assumptions around cell production / cambial activity (ON, OFF, or active with certain temperature+water- dependent rate). The end of the growing season is therefore also driven by the and earlier or later cessation of cambial activity. So the end of the growing season is unambiguously driven by cambial activity in the VS-model. Maybe you mean that it is ambiguous what climatic drivers are responsible for the end? Or do you simply mean that there is a non-linear behaviour with elevation? What would be really interesting- and what is within the scope of your model output –is related to my next comment.

Response: We reformulated the sentence to improve its clarity. Now it reads: *The extension was driven mainly by an earlier start of cambial activity, while the end of the growing season showed site-specific shifts over time, largely depending on climatic water balance.* (L. 187-189). Please, see also our following response and novel analysis on growth cessation and its climatic drivers.

L 179 „Due to amplified drought stress“- has no reference to any figure but is maybe the most interesting part. What – according to the model drives the onset and cessation? In my understanding Figure 4 a and b show the general growth deficits over the full season and we cannot see the drivers of start and stop for c). from Fig4 graph or S7. I think a more in-depth analysis on what environmental factor is 1) dominating or immediately active after growth start then 2) causes the cessation growth at the end of the season would be a more useful way to interpret figure 4 row c).

Response: Thank you for this stimulating comment. As already mentioned above, we assessed the frequency of zero integral growth rates for each calendar month together with its climatic drivers (cold X drought) to show their seasonal distribution and shifts with climate change. The novel approach is introduced in the *Material and Methods* section (L. 595-598), results are presented in the *Results* section (L. 189-192), and, most importantly, they are

visualized in the novel figure (**Fig. S10**). The results highlight a progressively declining frequency of growth cessation in winter (cessation due to cold limitation) but increasing frequency of growth cessation (drought-driven) during summer and autumn, reflecting the pace of climate change within SSP scenarios.

L 721 integrate legend on black blue and red line into the figure, maybe in an empty panel

Response: The figure was improved accordingly.

L190 From the figure you refer to here I am not convinced it is fair to say that the conifers look completely bimodal (i.e. similar height peaks with a strong troph between), and I don't think the Broadleaves can be described as completely non-bimodal.

This is what I see:

Broadleaves are largely *left*-skewed, in some cases with a shoulder or small secondary peak. Conifers are more strongly tending towards a bimodal growth pattern.

Additional thoughts: In the Figure S9, I would agree that overall there are bimodal features, but these are frequency distributions of all sites combined I assume. There, I would even say that Broadleaves also have bimodal features. How can these two results be consolidated?

Response: Please note that what is often called a 'bimodal' intra-annual growth pattern e.g., in the Mediterranean, is only rarely characterized by 'similar height peaks'. Instead, the spring peak is commonly substantially larger (at least +50%) than the autumn peak, primarily reflecting the seasonal moisture distribution (e.g., Camarero et al. 2010: *NewPhytol*; Pacheco et al. 2018: *STOTEN*; Campelo et al. 2018: *Dendrochronologia*; Vieira et al. 2022: *Forests*). The mean skewness of intra-annual integral growth rates increased from the baseline period (0.73) to 2080-2099 (0.93 under SSP5-8.5), suggesting a shift to right-skewed instead of left-skewed distribution. However, to tone down assessment of uni- and bi-modal features of intra-annual growth of conifers and broadleaves (that are subjective to some extent) and to consider change of our results after inclusion of new sites, we reformulated the sentence accordingly: *Consequently, the pattern of integral growth rates tended to shift into right-skewed unimodal or even bimodal form under high-emission SSP scenarios (Fig. S13). The nearly symmetric unimodal pattern persisted till the end of the 21st century only at humid sites of Picea sp. and Pinus sp., however, the latter genus being underrepresented in humid sites in our dataset. (L. 200-204)*

L217 change „decoupled“ with „diverged“

Response: Revised accordingly (**L. 256**)

L220 The results have not convinced me that these trends or decadal means of ring width deviations are unprecedented. I see two occasions where ring width deviations will go beyond the observed variability of the baseline period and that is for abies and piecea in the low lands for the two high emission scenarios.

Response: Thank you once again for this comment. As already described, we extended our analysis by forecasting growth during unusually cool-wet and warm-dry years within each forecast period. Our results show that although growth predicted for mean climate during future bi-decadal periods toward the end of the 21st century will mostly stay within the histor-

ical range, growth mainly under dry spells will be substantially beyond the range of baseline predictions (L. 205-229).

L 230 replace „by „ with „in“

Response: Revised accordingly (L. 267)

L239 ff: „onset“ not quite right, and „decoupling“ not right. An attempt: Indeed, high-emission SSP scenarios might cause continuously declining lowland forest growth and benefit (?) mountain forests by causing potentially fast turnover.

Response: Reformulated according to your suggestion. Now the sentence reads: *Indeed, high-emission SSP scenarios might cause continuously declining forest growth at dry sites and benefit humid forests with fast turnover.* (L. 277-279)

L 321 „models“ to „model“

Response: Revised accordingly (L. 411)

L 351: growth deficits are an inactivity? Methods say otherwise. Replace „inactivity“ with „relative reduction from optimal growth rate“?

Response: The sentence was reformulated to reflect our novel analysis on the frequency of growth cessation and its drivers for each calendar month. Now it reads: *Moreover, most of the simulated growth deficits were due to cold rather than drought stress, and the simulated growth cessation mostly occurred due to cold rather than drought-limitation.* (L. 303-305)

L 363ff It is unclear in these methods how the authors have combined the projection climate dataset and the historical climate data. It reads to me that (spatially interpolated) historical time series were obtained from climatological stations. The baseline period for the climate anomalies does not correspond to the baseline climate used for calibration of the VS-lite model. I wonder whether this causes problems with the simulations. It is further unclear whether statistical downscaling was performed on the low-resolution CMIP6 forcing data to accommodate high and low- latitude differences, which normally manifest at high-resolution, as realistically as possible.

Response: We improved the clarity of the text describing the climatic data and their processing (L. 475-481, 566-575). Since our study domain was expanded, we combined Czech gridded climatic data with European E-OBS dataset for other countries (considering the sensitivity test to the choice of the climatic dataset at Czech sites; L. 472-474). For climatic forecasts, we used bi-decadal anomalies instead of climatologies. Therefore, we were able to predict bi-decadal climatologies from the normal period at each site, including climatology for future mean, warm-dry, and wet-cool years. The original downscaling level of CMIP6 for SSP scenarios was $0.25^{\circ} \times 0.25^{\circ}$, which we consider sufficient given the use of anomalies instead of climatologies and bi-decadal resolution of our forecasts. We also extended Fig. S14 to present mean forecast bi-decadal climatologies, bi-decadal anomalies, and long-term climatic trends across our study domain.

L 422 with „the“ calibrated

Response: Revised accordingly (L. 532)

L 471: replace „by“ with „using“
Response: Revised accordingly (L. 593)

L 493 add „(Figure S9)“ somewhere
Response: Revised accordingly (L. 623; now referencing Fig. S13)

L 493: unclear where in the results this is reported. I only found a difference from the baseline in relative ring width in Figure 3, but nothing related to phenology and kinetics. Best highlight to the reader which figure you are referring to.

Response: We reformulated the sentence to improve its clarity. Moreover, we provide a reference to the figure showing the respective results. Now the sentence reads: *Finally, we divided the simulated tree-ring width for each forecast period, SSP scenario, and climatology by the mean simulated tree-ring width in the baseline period (Figs. 3, S6).* (L. 623-625)

L 496 ff I cannot find any tables or plots related to this analysis. Can you please better connect it to your results/supplementary materials?

Response: Results of the non-stationarity test are presented in **Fig. S4**. We shortly comment on them in the *Results* (L. 151-153) and discuss them in more detail in the chapter of *Discussion* on model limitations (L. 385-388).

L 703: needs explanation that the SSP symbols are also decadal averages (if they are).

Response: Caption of **Fig. 3** was revised accordingly.

Reviewer #3 (Remarks to the Author):

General comments

Tumajer and co-authors presented a manuscript on the radial growth of tree species in Central Europe. The paper is well-written, presenting a timely and interesting topic in the context of forest and climate change. It introduces a novel approach to calibrating a tree-growth model (VS-Lite) using remote sensing and dendrometer data. The innovative use of remote sensing for model calibration is particularly noteworthy given the increasing global availability of such products. The analysis are robust, and complemented with an online code that allows reproducibility. However, details of the methodological process, as currently described, might be difficult to comprehend for readers who are not familiar with such models and their approach.

Response: Thank you for your comments on our manuscript. To improve presentation of our approach, we shifted schematic **Fig. 6** (which we slightly modified from the original submission by differentiating between ‘growth deficits’ and ‘growth cessation’) into the appropriate section of the *Material and Methods*. Moreover, we improved captions of **Fig. S5** and **Table S3** to provide a direct definition of the model parameters.

Although the methods are solid, the suitability of this study for a journal such as Nature Communications needs consideration. The title and discussions suggest an analysis at a global or continental scale, since authors generalize their results for temperate forests. However, the analysis was only performed in sites within Czech Republic. This is certainly still valuable, but the way this is presented to the readers is misleading. No mention to the

study region is given until the Material and Methods (line 336). A map of the sites is somewhat hidden among the last figures in the supplementary. Results are therefore not fully representative of Central Europe as a whole, and neither for temperate forests, as the title and abstract suggest. Incorporating data from the Alps and other lowlands in Central Europe could broaden the elevation range. If this is not possible, it should be made clear from the beginning of the manuscript what are the boundaries of the input data, and generalizations of the results must be amended. If that's the case, also the title should reflect the geographical limitation of the study region.

Response: Thank you for your thoughtful comment. To improve the representativeness of our dataset, we extended the network of tree-ring chronologies for about >1,000 new sites from the RemoteForest database. Accordingly, the extended dataset includes 2,013 chronologies of 15 species across 12 countries distributed in Central (CZE, GER, AUT, POL, SVK), Eastern (UKR), and Southeastern (ROM, BUL, SLO, CRO, BIH, ALB) Europe. Therefore, these sites adequately represent the temperate forest biome and partially other forest types (subalpine forests, Mediterranean forests). We also included a map of the study domain as **Fig. 1**, intended to be placed at the end of the *Introduction*, to provide an overview of our study area before presenting the main results. The figure shows the spatial distribution of our sites (Fig. 1a) but also the comparison of their climatic conditions with the entire climatic niche occupied by given species in Europe (Fig. 1b). Finally, we carefully reformulated statements about the study area in the entire manuscript (e.g., **L. 362-367**) to better characterize spatial representativeness of our network of sites (including adjustments of the manuscript title).

There is also a question of whether a monthly-resolution model is sufficiently detailed, as minimal effects of climate change until 2050 may result from this coarse time resolution.

Response: We agree that phenological shifts under limited rates of climate warming might be subtle to be fully captured with a monthly-resolved VS-Lite model. Accordingly, we included a new chapter to the *Discussion* where we elaborate on the limitations of our dataset and model, considering also effects of coarse temporal resolution (**L. 394-409**). However, the mean rate of growing season extension predicted by the end of the 21st century under high-emission scenarios, 0.39 days per year (≈ 1.44 months over 110 years), is close to currently observed phenological shifts in Central Europe. This confirms previous studies showing a reasonable performance of the VS-Lite for estimating growth phenology (Seftigen et al. 2018: *Global Ecology and Biogeog.*).

Additionally, there should be a discussion on how relevant growth reactions to average future climate conditions are, given that forest growth appears more influenced by climatic extremes (e.g., sudden drought spells, heat waves, subsequent dry years) rather than chronical temperature increase as indicated by climate projections.

Response: Thank you for this stimulating comment. To extend our analysis, we used the VS-Lite to predict tree-ring widths and intra-annual growth patterns for two types of meteorological extremes (cool-wet years and warm-dry spells) toward the end of the 21st century. Our results show that growth reductions caused by meteorological extremes will go far beyond the historical variability, mainly considering dry sites and high-emission SSP scenarios (see. e.g., **Figs. 3-5**). We present the impacts of forecast extremes on tree growth in the novel subchapter of the *Results* (**L. 205-229**).

More generally, the paper is missing of a discussion paragraph about the main limitations of the study (e.g., limited representativeness of the sites for Central Europe, intrinsic model limitations, tree mortality not modelled, climate extremes).

Response: We included a new sub-chapter of the *Discussion* section stressing the main limitations of the study, mainly concerning (i) representativeness of the tree-ring dataset, (ii) VS-Lite performance, and (iii) limitations due to bi-decadal resolution of climatic forecasts (L. 361-409).

Furthermore, I have some doubts about the robustness of the dendrometer data used for calibration. The calibration sites (57 sites) are not shown and little details about them are given in the manuscript. Therefore their representativeness for the environmental gradient cannot be assessed. Additionally, most dendrometer data only cover one year, which may not represent the historical climate period. This should be clarified by the authors.

Response: We produced a new map showing the distribution and temporal span of available dendrometer data (Fig. S15). We agree that the availability of the dendrometer data and their temporal span is limited. Therefore, (i) we did not use them for model calibration but rather independent validation at a subset of sites, and (ii) performed additional validation using NDVI, which was available for all sites and longer periods. Nevertheless, we consider the use of dendrometer data – although limited to a few sites and years – as an important complementary step for validating VS-Lite intra-annual simulations which has not been frequently performed in previous studies. We further revised the text and improved our statements about the dendrometer dataset to avoid confusion about its replication (e.g., L. 548-549).

I provide here below a list of more specific comments that can help to improve the manuscript.

Specific comments

L.46 “future CO2 emissions”. Although this is correct, it sound like the model uses CO2 time series, but this is not the case. Perhaps write just “future climate change”.

Response: Revised accordingly (L. 46).

L. 50-53 “mountain forests” and “lowland genera”. The title of the paper suggests a focus on lowland forest but the abstract describe differences between mountain and lowlands. This is confusing. It is also not clear at this stage what the author mean by lowland genera and how they are differentiated.

Response: With expansion of our dataset toward Eastern and Southeastern Europe, we cluster our sites according to climatic water balance instead of site elevation. We further reformulated the sentence to improve its clarity. Now the sentence reads: *During the second half of the 21st century, high-emission scenarios predicted growth acceleration in humid coniferous forests due to growing season extension and enhanced growth rate. In contrast, the extension of the growing season was insufficient to compensate for declining summer growth rates at drier sites, resulting in significant growth reduction for all genera, particularly during dry years.* (L. 49-53).

L. 56 “sustainable”. Not self-explanatory. Not really clear at this stage what does it mean “sustainable” in terms of stem growth. Please clarify.

Response: The statement was reformulated. Now the sentence reads: *Furthermore, we highlight that only low-emission scenarios support non-declining stem growth in dry forests with current species composition.* (L. 55-57).

L. 84 “negative trends” I am unsure if “trends” is the correct term to be used here. Do you mean negligible increase in total growth even if timing of growth onset is anticipated? Please clarify.

Response: The sentence was reformulated. Now it reads: *For instance, an earlier start of the growing season might result in no change or even narrower tree rings of temperate tree species.* (L. 91-93).

L.96 “instant” What do you mean by “instant”?

Response: We replaced instant with *monthly* to clarify the meaning of the sentence. (L. 105).

L. 96 Figure 1. Overall, I like this comprehensive figures about the functionality of VS-lite but it might be difficult for readers who are not familiar with the model to understand all parameters and functions. First of all, you referred to a modified version of the model, and at this stage the model is not described yet (i.e., what has been modified? Modifications were not described yet). Furthermore, when at the end of the caption you describe the parameters calibrated at each site, sites and study design it has not been described yet. This is confusing. Additionally, in Fig.S8 it is written that parameters explanation is reported in Figure 1. Parameters are displayed, but not explained. For readers who are not familiar with the model, this is hard to understand. Please clarify.

Response: The schematic figure of the VS-Lite mechanisms was moved to the *Material and Methods* section (Fig. 6), i.e., at the relevant part of the manuscript where the model functionality including our modifications is introduced. We also included an additional column in Table S3 and footnotes to Fig. S5 with a description of the individual parameters of the model.

L. 105 “ dryer lowlands to the alpine treelines”. When I read this I expect this study to cover a gradient from the Mediterranean to the upper treelines in the Alps. However when I look at figure 3 I realize that the maximum elevation range is 1490 meters. I am not sure if this can be considered the upper tree line in continental Europe. Please provide the elevation range and a description of the study sites/region.

Response: As already mentioned above, we extended the spatial gradient covered by our dataset in a revised version of the manuscript by including new sites from Eastern and Southeastern Europe. We also characterized environmental and geographical gradients covered by our sites in the chapter about the study area. The respective section reads: *Our sites representatively cover mixed forests in Central, Eastern, and Southeastern Europe across prominent latitudinal (42.4-51.8° N), longitudinal (10.6-25.5° E), and elevational gradients (153-1713 m). The mean annual climatic water balance of individual sites defined as the difference between annual precipitation and potential evapotranspiration during the 1961-2020 period varies between -160 mm and +1230 mm. The dataset covers the central*

areas of the current climatic niche of the main genera but sites similar to dry-cool (*Picea* sp., *Pinus* sp.) or warm (*Fagus* sp., *Abies* sp.) distribution margins in Europe remain underrepresented (Fig. 1b) (L. 444-452)

L. 106-107 “83 intra-annual series of radial growth”. Also this is misleading. By reading this I expect the authors to have used 83 sites with longer time series observation reflecting mean intra-annual growth of the species as a representative for an historical period. Instead, only 56 sites were used and mostly with only one year of observation. Please rephrase it clearly. This also refers to my general concern reported above about dendrometer data.

Response: The sentence was reformulated to improve clarity. Now it reads: *To independently validate simulated growth phenology and kinetics, we compared the simulations with dendrometer data available mostly for a single year at 57 sites, and with the proxy of the leaf phenology estimated from the Normalized Difference Vegetation Index (NDVI) for the 2000-2020 period at each site (L. 115-118).* Moreover, we included novel Fig. S15 presenting spatial distribution and temporal span of individual dendrometer sites.

L. 117 Besides a positive effect on radial growth, an earlier start of the growing season could also lead to a higher vulnerability to late frosts with low growth rates in years with frost damages. I believe this is not considered VS-lite due to its monthly resolution. Please acknowledge this aspect (somewhere in the manuscript).

Response: We mentioned increasing risks of late frosts with climate warming in the *Introduction*: *Notably, the effect of changing spring temperature on temperate growth phenology and how it translates into tree-ring width is not linear due to the interplay between chilling and forcing temperatures driving spring cambial reactivation, and increasing risks of late frosts with climate warming (L. 84-87).* Moreover, we discussed late frosts as one of the limitations of the VS-Lite model in a subchapter of the *Discussion* focused on the limitations of our approach: *Further studies should test improvements in phenological mechanisms of the VS-Lite model, including effects of winter conditions on growth phenology through chilling-forcing interactions, risks of late-frost damage after cambial reactivation, or non-climatic drivers of growth phenology. (L. 402-406).*

L. 117-121 “systematic variability” Not really clear to me what it is meant by systematic variability, whether it is referred to only one variable at a time or if it includes all the variables mentioned subsequently. Please provide a more precise explanation of this term and indicate whether it applies to each variable individually or collectively.

Response: We simplified the sentence. Now it reads: *Moreover, we expected that the net balance between the positive effects of growing season extension and negative effects of summer growth decline would vary (ii) along the climatic gradient, (iii) among SSP scenarios, and (iv) during future meteorological extremes including cool-wet and warm-dry years. (L. 127-130).*

L. 125-126 “observed and simulated tree ring width chronologies” For the calibration period 1961-2020?

Response: The sentence was reformulated. Now it reads: *During the 1961–2020 calibration period, the mean (\pm standard error) correlation coefficient between observed and simulated tree-ring width chronologies was 0.42 (\pm 0.003). A significant correlation was found at 84 %*

($p < 0.05$) and 66 % ($p < 0.01$) of 2,013 sites for which we calibrated the VS-Lite model (L. 137-140).

L. 127 Figure 2. When I look at these results, so far I still have no idea about the geographical area of reference. I can only assume that the authors used chronologies from all around the world. This referred to my main concern about the authors not being very clear about the geographical constraints of their application. Being more transparent about the geographic limits of the study would help the interpretation of this figure.

Response: Please, see our responses above about how we improved the presentation of the study area. We included new sites to our network. Moreover, we placed the map with a comparison of our sites with full species climatic niche within Europe at the end of the *Introduction* (Fig. 1). Finally, we reformulated specific parts of the manuscript (e.g., L. 362-367), including the title, to better reference the spatial representativeness of our dataset.

Other suggestion: consider adding a legend for the color lines in the figure and not only in the caption description. Quicker to interpret.

Response: Fig.2 was enhanced according to your suggestion.

L. 129-131 there is also higher peak of growth rate for all species except *Abies alba* in dendrometer compared to model data (which is then offset by higher values for model data in autumn). Having just one year of dendrometer data for most sites (according to Fig. S2) could have quite a big influence on the comparison of dendrometer and model data. Please refer to my general concern mentioned above.

Response: As mentioned above, we reformulated specific parts of the manuscript referring to dendrometer data and included novel map with their distribution (Fig. S15) so as not to make false suggestions about their replication. Moreover, we discussed deviations between VS-Lite and dendrometer data apparent in Fig. 2b in the novel chapter of the *Discussion* focusing on model limitations: *The simulated phenological shifts might have been affected by the monthly temporal resolution of the VS-Lite model. For instance, autumn integral growth rates were slightly overestimated compared with dendrometer records (Fig. 2b). Wood formation models operating on daily temporal resolution might be better suited to simulate summer growth quiescence and short-term autumn reactivation under seasonally dry conditions.* (L. 394-398).

L. 137. “diffuse stands or below-canopy genus” Unclear. I would not generally agree with the characterization of *Abies* sp. (I guess mainly *A. alba*) as below-canopy species, but this probably depends on the specific forest types and stand structure. Perhaps rephrase it into “shade intolerant” and “shade tolerant”.

Response: *Abies* sp. was dropped from the sentence because this genus shows improved agreement between simulated intra-annual growth patterns and NDVI after including chronologies from the Carpathians and Southeastern Europe (Fig. S3). This is probably due to the sparse extent of *Abies alba* in the Czech Republic, where it commonly represents admixed species of stands dominated by *Picea abies* or *Fagus sylvatica*, but close-to-monospecific *A. alba* stands in remote areas of Carpathians.

L. 140 Figure S5. I believe that caption in figure S5 would be the right place to fully describe the meaning of the temperature and soil moisture parameter. Please also explain what the other two parameters are. I believe Acor means autocorrelation but this is not clearly explained.

Response: The caption of **Fig. S5** was enhanced according to your suggestion.

L.142 Figure 3. Upon examining figure 3, it is evident that the elevation gradient of the sites is not representative of the alpine region. Instead, it only reflects central Europe from lowlands to mountain elevation belts. The primary conifer species covering subalpine to tree line elevational zones appear to be absent, such as the Swiss stone pine.

Response: The elevation gradient of the extended dataset is 153-1713 m (**L. 447**), which is representative of montane and alpine regions of Hercynian mountains, Carpathians, and most Balkan Peninsula mountain ranges, including sites within local treeline ecotones mainly in the Carpathians. Accordingly, there is sufficient replication in the most humid category of sites (mean annual climatic water balance > +700 mm; presumably highest elevations) for *Picea*, *Abies*, and *Fagus* species. Our extended dataset included only 3 sites of *Pinus cembra* due to its relatively sporadic occurrence in the Carpathians compared with the Alps.

Specific comments: 1) what does the color line indicate. If colors are not really used (species are already differentiated by panels) wouldn't it be more efficient to use color lines to differentiate elevation belts and optimize the space? Some boxes are blank (not ideal). By optimizing the figure in this way, size of the panels could be increased enhancing interpretability (i.e., printed on half-page, symbols can be hardly differentiated).

Response: The number of blank boxes was reduced with the extension of our dataset. Accordingly, we preferred to keep the original layout of the chart. Furthermore, the figure was extended by incorporating forecasts of tree-ring widths under cool-wet and dry-warm extremes at the end of the 21st century into the right part of each panel. We aim to use colors for individual genera consistently through all figures in the manuscript to visually aid their reading.

2) I suggest adding an ab-line or a shadowed area in gray to differentiate calibration vs forecasted period;

Response: We modified the chart according to your suggestions.

3) shadowed area within the graph and the color used for the species is barely visible, please enhance.

Response: We removed the shadowed area originally used to highlight a range between SSP scenarios. Instead, we used shadowed areas on the right edge of panels to visually separate simulated tree-ring widths based on warm-dry and cool-wet years from the simulations based on mean climatologies.

L. 143-145. I do not understand very well this sentence. Please rephrase.

Response: This sentence was dropped.

L. 153 "neglectable" Do you mean "negligible"?

Response: Statement was revised accordingly (**L. 167**).

L. 147-149 and L.162 Confusion when referring to low, middle, and high elevations. Please provide numbers about the elevational gradients and correct (throughout the manuscript) the term “treeline”, which cannot be used as higher site is at 1490 m a.s.l. (unless in Czechia the natural upper treeline occurs at these elevations, but I doubt this is the case).

Response: Since the geographical range of the dataset was extended, we clustered sites according to climatic water balance instead of elevation belts. We use the terms dry (annual climatic water balance -160-+200 mm), moderate (200-700 mm), and humid (700-1230 mm) for individual clusters. The definition of these clusters is provided at **L. 160-165** and **L. 612-617**. We also corrected the use of the term *treeline* in the manuscript.

L. 170 Figure S7. The figure is really hard to interpret. Panels are very small. Consider splitting into two separate figures. Additionally, the fourth panel for scenario SSP5-8.5 seem a repetition of what already shown in Figure 4.

Response: We split the chart into two separate figures and dropped the redundant plate for SSP5-8.5. (**Figs. S8, S9**).

L. 172 Figure 4. Please explain better how to read the phenology (c) graph. It took me a while to understand why there are two trends lines in the panels because it is not clear which line represents the onset and offset of the growing season. Please provide a clearer description in the caption.

Response: The figure was restructured to be more comprehensive by splitting phenology into two separate rows for the start and end of the growing season.

L. 177. What do you mean by ambiguous trend? Please clarify, perhaps describing this more directly in the caption. Also, trends are difficult to detect from figure S7 (see two comments above).

Response: The sentence was reformulated to improve clarity. Now it reads: *The extension was driven mainly by an earlier start of cambial activity, while the end of the growing season showed site-specific shifts over time, largely depending on climatic water balance* (**L. 187-189**).

L. 182 Figure 5 and Figure S8. Figure S8 is suitable for supplementary material, but Figure 5 should be enhanced to better illustrate the differences between climate change scenarios. For instance, a reduced number of examples can be presented (as the elevation differences are often negligible), and varying line colors or types could be employed to emphasize the distinctions between climate change scenarios within the same panel. In Figure S8 there are panels that repeat results from Figure 5 (necessary?).

Response: We extended figures by including simulated monthly integral growth rates for warm-dry and cool-wet years for the 2080-2099 (indicated by line types). We prefer to show all four scenarios in **Fig. S11** because it shows rare genera in addition to **Fig. 5**.

L. 195 “cambial activity in the region”. Once more, I have no idea about the region (whole Central Europe?). I doubt whether Materials and Methods were simply shifted to the end of the manuscript after having initially structured the manuscript in the classic IMRaD format.

Response: Please, see our responses above about how we improved the presentation of the study area by including new sites in our network. Moreover, we placed the map of site distribution with comparison to the full climatic niche of the main species at the end of the *Introduction* (**Fig. 1**). Finally, we reformulated specific parts of the manuscript (e.g., **L. 362-367**) including the title to better reference the spatial representativeness of our dataset.

L. 209-210. This text and Figure S.11 indicates that temperature and precipitation differences are minimal between the four scenarios in the 2050s. Interestingly, in the 2090s, differences in precipitation between scenarios appear greater than those in temperature. However, as the figure shows yearly courses, this might be misleading. A figure showing deviations from the 1961-2020 means (standardized by standard deviations) could help estimate these differences. Additionally, a line plot of temperature trends from 1960 to 2100 would clarify developments. Furthermore, Figure S11 could be enhanced by incorporating annual time series or trendlines to more effectively illustrate the rising temperature trends and variations in precipitation over time, together with panels with intra-annual pattern, without necessarily repeating the baseline for two different periods as it is currently done (confusing).

Response: We extended **Fig. S14** by incorporating two new plates. The extended figure presents forecast bi-decadal climatologies (a), bi-decadal anomalies from the normal mean (c), and long-term trends in mean annual temperature and annual precipitation (b).

L. 215-216 “if climatic extremes, including drought spells, (...)”. The recent past especially in Central Europe showed that tree growth and forest vitality is mainly determined by climatic extremes. This limitation should be discussed more extensively (see generic comments).

Response: Thank you for this stimulating comment. As already outlined, we simulated the potential impacts of extreme climatic events by forecasting annual and intra-annual growth for mean, extremely warm-dry, and extremely cool-wet years toward the end of the 21st century (e.g., **L. 205-229, Fig. 3-6**).

L. 225 “populations” In forest genetics, populations usually denotes specific adaptations and genotypes. I do not think that is what you mean here. Please clarify.

Response: We used the term *forests* instead of populations (**L. 264-266**).

L. 228-229 “long term growth reductions often precede forest die off”. Agree. This study can only speculate on that issue because the model does not simulate tree mortality. This aspect should be discussed more clearly in the manuscript.

Response: We agree that the VS-Lite is neither capable nor intended for simulating tree mortality. However, our simulations pointed out a trend toward a significant reduction of tree-ring widths during future dry years, which might indicate an increasing risk of forest dieback. Therefore, combining wood formation models with models of stand dynamics is a promising future research avenue to simulate links between declining growth rate at the tree level and mortality at the stand level, i.e. it could be used to investigate early-warning signals of tree death. We discussed this accordingly: *Integrating tree-ring formation models, such as VS-Lite, with models simulating stand processes is essential for understanding how projected growth reduction contributes to the risk of tree die-off* (**L. 287-289**).

L. 259-260 “indirect climatic effects” unclear what is meant here by indirect climatic effects. Do you intend, for example, natural disturbances?

Response: We reformulated the sentence to improve its clarity. Now it reads: *Further studies should test improvements in phenological mechanisms of the VS-Lite model, including effects of winter conditions on growth phenology through chilling-forcing interactions, risks of late-frost damage after cambial reactivation, or non-climatic drivers of growth phenology (L. 402-406).*

L. 274-284. I did not find this paragraph very informative, at least in the way it is phrased now. The main question appears to be: will central European species known adapted to this pattern be able to tolerate these repeated reduction or will this lead to tree mortality? I suggest expanding a bit this aspect which is very interesting but in relation with recent studies from literature on tree mortality.

Response: The paragraph was rephrased to improve its clarity and extended considering novel findings based on simulations for extremely warm-dry and cool-wet years, i.e., risk of disappearance of autumn growth period during future warm-dry years at dry sites (L. 328-336). Moreover, we presented a paragraph addressing the central question, i.e., will temperate species be able to adapt their growth patterns to seasonal moisture availability, considering reports on growth bimodality from dendrometers during dry years and occurrence of latewood IADFs in Central Europe (L. 337-360).

L. 289-291. I am a bit puzzled about this sentence. To my knowledge, the distinction between species with isohydric and anisohydric strategies is not equal between broad leaves and conifers. There are some conifers that are more drought resistant than broad leaves. Please clarify, expand or rephrase.

Response: This part has been dropped.

L. 331-333 “mechanistic understanding or ecosystem responses to future growth dynamics is needed” agree, however this has been addressed in many studies using process based forest models. Please expand a bit more this discussion in the context of recent literature, in particular giving some examples of applications with process based models on forest growth and tree mortality.

Response: We included the following paragraph at the end of the *Discussion*, stressing mainly the potential for integrating wood formation and stand models: *Various stand models provided vital information about the responses of European temperate forests to environmental drivers and their future changes. However, some key components of their predictions, including mortality rates, remain considerably uncertain. Given the empirical link between declining growth trends and increased mortality, integrating forecasts of growth phenology and kinetics by wood formation models into stand models might improve the mechanistic understanding of ecosystem responses to future shifting seasonality of temperate forests. (L. 421-427)*

L. 348-352. To what extent do you remove long-term growth-trends which might be a reaction to climate change in the past decade by using a spline detrending compared to a more conservative detrending or using BAI? Please clarify.

Response: We preferred to avoid using BAI due to the low mean age of trees in our dataset (128 years) and a significant proportion of trees younger than 100 years, i.e., within cohorts with ontogenetic trends persisting more in BAI than in TRWi (Klesse and Bigler 2025: Dendrochronologia). Moreover, metadata on DBH and pith offset were missing for most of our cores, leading to less precise estimates of BAI. Spline is the preferred method in network studies employing VS-Lite models for growth forecasting (Dannenberg 2021: ERL; Sanchez-Salguero et al. 2017: GCB; Camarero et al. 2021: GCB; Seftingen et al. 2018: GEB). Moreover, we added an explanation of our choice of spline length at 60 years into the revised manuscript: *The spline length was chosen to retain half of the variability within the calibration period (1961–2020) as a signal in the detrended series while removing the other half as an ontogenetic trend. The assumption that half of the variation in tree-ring widths is attributable to tree aging is justified by the average age of our sampled trees, which is 128 years.* (L. 455-459)

L. 364 “totals” What does “totals” mean?

Response: Reformulated to *monthly total precipitation* (L. 475).

L. 371-383 This section describes the IPCC climate change scenarios, but lacks details on how GCM projections were downscaled to the site level. Additionally, it mentions climate anomalies by decade, though the VS-lite model requires monthly time series. More information on the processing and preparation of climate data is needed.

Response: We clarified the text describing the climatic data and their processing (L. 475-481, 564-577). For climatic forecasts, we used bi-decadal anomalies, i.e., forecast differences between future and historical climatic variables for each calendar month. Therefore, we were able to calculate bi-decadal climatologies, i.e., forecast monthly values of climatic variables at each site, including climatology for mean, warm-dry, and wet-cool years. The level of downscaling of SSP scenarios was $0.25^\circ \times 0.25^\circ$, which we consider sufficient given the use of anomalies instead of climatologies and bi-decadal resolution of our forecasts. To streamline the text, some details about the SSP scenarios were moved to **Table S2**.

L. 381-383. “we refer to SSP1-2.6 (...)” This sentence is essential for interpreting figures and results but it is mentioned only at this late stage in the manuscript. Once more, it makes me doubting that the manuscript was initially structured in IMRaD format without a final check for consistency. understanding of the results. Please place it earlier in the text.

Response: We checked the flow of the manuscript for clarity and defined low-emission and high-emission scenarios also in the *Results* section, i.e., at the first use of these terms (L. 160-170).

L. 387-388. Recent studies based on dendrometer data suggest a strong influence of VPD on radial increment/growth. Should this also be accounted for? S_r is the monthly resolution of the model to coarse for it? Please discuss these aspects. Also, there is an "a" is missing between “in” and “specific” in line 387.

Response: We discussed aspects of implementing VPD into Vaganov-Shashkin and VS-Lite models in the *Discussion* section: *Finally, VS-Lite model might benefit from incorporating additional drivers of wood formation like vapor pressure deficit⁴³ which was has recently*

been successfully implemented in the daily Vaganov-Shashkin model⁵⁶, although its predictive power for growth at a monthly scale still needs to be tested. (L. 406-409).

L 406-407 The reference should probably be to Fig. 1a or 1c instead of 1b.

Response: Thank you for pointing this out, we corrected the reference (now it is **Fig. 6a**).

L. 429 “83 growing seasons at 57 sites”. Only at this stage of the manuscript it is clear to me that calibration sites with dendrometers data are 57. However, it is still unknown whether these sites cover the environmental range of variability across the study region. I would suggest to move the map, currently in the supplementary, in the main manuscript by adding also the location of the these sites.

Response: We produced a new map showing the distribution and temporal span of available dendrometer data (**Fig. S15**). We also reviewed the entire manuscript and clarified the presentation of the limited representativeness of the dendrometer dataset (e.g. **L. 548-549**).

Table S1. What are species “Q.sp.” in *Quercus* and “F. sp.” In *Fraxinus*? If species is unknown or hybrid/exotic, why was it not excluded from the analysis?

Response: Since our dataset of tree-ring width chronologies is based on database records, sp. refers to sites whose metadata were specified only to the genus. Currently, this holds for a few sites of *Quercus*, *Fagus*, and *Acer* genera. We explained the meaning of the sp. suffix in the table footnote.

Reviewer #3 (Remarks on code availability):

I only gave a check to the scripts and code and it appears all on order and reproducible. Upon publication, I suggest archiving the code in a long-term official repository (e.g., Zenodo, Dryad).

Response: We plan to link the GitHub repository with Zenodo and generate a doi in case of the manuscript acceptance. Accordingly, a link with doi instead of GitHub will be provided in the manuscript in case of acceptance.

Reviewer #4 (Remarks to the Author + Remarks on code availability):

I reviewed the R data repository and it appears that all the necessary inputs to run the code are provided. However, I didn't run the code or examine the inputs closely due to time constraints.

Response: Thank you for your support to ECRs with their review.

Reviewer #3 (Remarks to the Author):

I truly appreciate the efforts by the authors to intensely revise the manuscript. All comments and recommendations were thoroughly addressed and I believe that the manuscript has been greatly improved. I particularly appreciate the honesty in acknowledging the geographical constraints of the study by providing a clear distribution map of the dataset and by revising accordingly the title. Furthermore, extending the dataset to enhance its geographical representativeness and expanding the simulation analysis to include meteorological extremes provided additional emphasis to the results.

Response: Thank you very much for the positive evaluation of our responses and for reviewing our manuscript again. Please, see our responses to your additional comments below.

I have checked each response made to each of my comments and I am satisfied the new version. I only have a few points that need to be addressed, among them how the two alternative climatologies were derived (cool-wet, warm-dry). This is explained only from line 569 (section Growth forecasting). The two alternative climatologies were derived by assuming a reduction in temperature and precipitation from climate projections. Please move this to the "Climatic data and climatic scenarios" section and refer to it earlier in the manuscript. This adjustment is important as these assumptions affect the simulated outcomes and their interpretation.

Response: Thank you for pointing out that the concept of alternative climatologies might not be clear to the reader until they reach the Material and Methods section. We adjusted the last paragraph of the Introduction to clarify the definition of mean, cool-wet, and warm-dry climatologies and to justify our motivation to include them in our modeling study. The specific part reads: *"To simulate the effects of continuous climate change and account for potential shifts in the intensity of meteorological extremes, we used SSP scenarios to define intra-annual climatologies representing mean, extremely warm-dry, and extremely cool-wet years expected during four bi-decadal periods through the 21st century (see the Material and Methods section for details on how the climatologies were derived)."* (L. 121-126)

Moreover, we moved the detailed definition of alternative climatologies to the "Climatic data and climatic scenarios" subchapter of the Discussion.

I provide some specific comments to the revised version here below:

L.124-125 Please add a brief motivation for adding these two alternative scenarios. Otherwise comes as a surprise for the reader. You could also add "(see details on how scenarios were derived in the Methods section)".

Response: Please, see our previous response on how we described alternative climatologies in the last paragraph of the Introduction.

L.151 "diffuse stands" I suggest using the term "sparse". Already commented in previous version, diffuse is ambiguous.

Response: The statement was revised accordingly (L. 157)

L.155 I think you could simply title is “Growth forecast under climate change projections” as these are the standard projections provided by climate models. Then in the text below you clearly acknowledge that in the manuscript you referred to it as “mean climatology”

Response: The title was revised accordingly (L. 161)

L.223 “Models suggested (...)” If you referred to fitted models to the results, I suggest simply writing “Results suggested (...)”, to not confuse with model simulations.

Response: The statement was revised accordingly (L. 229)

L.421-427 Referring to my previous comment (“Please expand a bit more this discussion about process based forests models”). I did not mean to discuss the need to integrate wood formation into stand models. Models can always be made more complex, but integrating fine-grain processes such as wood formation – and therefore more and more parameters - into e.g., stand or landscape forest model, this does not always guarantee that the models would work better for modelling processes. I meant to acknowledge more clearly the limitations of models like VS-Lite in the context of a broader literature on forest process-based modelling. Using a model that includes stress-induced mortality due to climate and disturbances on the same sites, will likely provide a much more dramatic picture than what currently presented in this study. Please add a short paragraph at the end of the Model and database limitations acknowledging these aspects correlated with recent literature (e.g., see Bugmann et al 2019 Ecosphere).

Response: We extended the subchapter about the VS-Lite model limitations accordingly. The respective part reads: “*Finally, the empirical model of tree-ring growth used in our study captured non-linear but fairly continuous shifts in phenology and intra-annual growth patterns in a response to changing climate. However, sites where climate change will trigger stochastic events including increased mortality might experience more dramatic and abrupt shifts of both intra-annual and inter-annual growth patterns. Accordingly, our forecasts of intra-annual growth at the tree-ring level may be less reliable at sites affected by stand-replacing forest disturbances in the future.*” (L. 404-410)

L.475 “we forecast” You did not forecast but “retrieved climate change projections from xx-yy database”

Response: The statement was revised accordingly (L. 472)

L.480 “Climate Change Knowledge Portal” Please add a reference or link to the resource.

Response: The link was provided at the respective place in the text (L. 478). Moreover, we list all links to relevant data sources in the Data availability statement (L. 808-822).

L.492 I suggest adding here the description on how extreme climatologies were derived.

Response: The description of extreme climatologies was moved to the indicated subchapter of the Discussion.

Reviewer #4 (Remarks to the Author):

Response: Thank you for your support to ECRs with their review.

Reviewer #5 (Remarks to the Author):

Task: The editor asked me to evaluate the authors' responses to the comments from the initial reviewers. I was not asked to provide an independent review of the revised manuscript. I was also asked to specifically focus on the response to comments from Reviewer #1 (R1) and Reviewer #2 (R2), since neither was available to assess the authors' responses to their comments. To complete this task, I read the revised manuscript and the authors' responses to the reviewers. I focused my evaluation on the major issues.

Summary: Overall, the revised manuscript addressed the major concerns of the reviewers. To do so, the authors expanded their focus with additional data and conducted additional analyses. However, the manuscript would still benefit from a more thorough explanation of its statistical framework and more design attention to figures.

Response: Thank you for evaluating our response letter and reviewing our manuscript. Please, see our responses to your comments detailed below.

Evaluation of major concerns:

1. Major shared concern: Not appropriate for the journal

R1 thought the paper lacked sufficient novelty

- Response: The authors added new components that expanded prior work including new analyses exploring potential tree growth responses to future climatic extremes and expanded their literature to address the importance of understanding differences in response among species.

R2 thought the paper had too much of a regional focus.

- Response: Authors expanded their dataset to new sites that expanded the spatial inference to sites in Eastern Europe and the Balkans and included more sites with broadleaved tree species.

Evaluation: Authors made a good faith effort to address the concerns.

Response: Thank you for your positive evaluation of our response.

2. Major shared concern: Difficulty in making sense of the results

R1 noted several specific instances where they had trouble following the methods and results.

R2 noted in their general comments that the results were hard to interpret in part because of a lack of clarity in the methods.

Evaluation: I read the revision de novo. With some careful reading, I could track how the results were obtained from the methods. However, it is challenging to follow the results without first reading the full Method and Material section.

Response: Thank you for pointing out challenges when reading the Results before the Material and Methods section. We extended the description of our methodology—particularly the definition of mean and extreme climatologies of future climate change—in the last paragraph of the Introduction to improve the comprehensibility of our study for those who read the Results section before the Materials and Methods (L. 121-126).

Also, the graphics are dense and not always sufficiently explained. For example, Fig 1b – climate variables on axes not entirely explained. Not sure why both r and R² were presented in Fig. 2 – seems redundant. And splitting out the fits for the different datasets also seemed like an unnecessary detail.

Response: We modified figures accordingly, i.e., extended variable definition on axes in Fig. 1b, dropped R² from Fig. 2a, and dropped dataset distinction.

3. R1 identified concerns about the meaning of significant differences.

Evaluation: The statistical analyses used in this paper are not fully described and the responses to the R1 comments did not clarify the situation. I am not saying the stats were inappropriate, but the choice of metrics (t-tests, GAMs) is not justified nor is sufficient analytical details provided.

Response: We described individual statistics and justified their use in the Material and Methods section.

- For GAM, it reads: “*We preferred generalized additive models for this purpose considering their flexibility towards potentially non-linear shifts in intra-annual growth with climatic water balance.*” (L. 605-607).
- For the t-test, it reads: “*For both applications of the Student’s t-test (i.e., monthly integral growth rates and annual simulated tree-ring widths), we used the one-sample, two-tailed version with heterogenous variance (Welch approximation) to test whether forecast for a given bi-decadal period significantly differs from the average of the baseline period (1961-2020).*” (L. 629-633).

4. R2 had questions about model calibration and validation. The main concern was using annual ring widths rather than monthly growth to calibrate the VS-lite model.

Evaluation: Authors noted that intra-annual data from dendrometers was ideal but such data was not available at the scale of the study. Their alternative – calibrate with annual and validate with intra-annual – struck me as a reasonable compromise. That said, a more thorough

evaluation of the validation metrics (Fig. 2b) is required. How good is good enough for the questions they are asking? There is a timing difference and maximum growth difference.

Response: Thank you for this comment. In addition to a visual comparison between dendrometer data and simulated intra-annual patterns of integral growth rates, we added numerical statistics to quantify their agreement: “*We also calculated Pearson correlation coefficient, root mean squared error, and temporal offset of seasonal growth peaks (i.e., months with highest growth rate within the year) to statistically evaluate coherence between integral growth rates and dendrometer data.*” (L. 557-560). The values of these statistics have been included directly in **Figures 2b, S1, and S2**.

5. R2 expressed concerns about the performance of the VS-lite model and its ability to predict growth of hardwood species.

Evaluation: Revision included the details and caveats requested.

Response: Thank you for your positive evaluation of our response.

6. R2 wanted more details on how drought-effects during the summer were quantified. This question was related to the distinction between growth onset and cessation.

Evaluation: Revision included a new analysis and results (Fig. S10) that addressed these concerns.

Response: Thank you for your positive evaluation of our response.